# Validity of Online Patient Medication Reviews and Ratings (PMRRs) for Treatment Satisfaction with Medication Therapy Among Older Adults with Antihypertensive Medications

**DOI:** 10.3390/healthcare13222918

**Published:** 2025-11-14

**Authors:** Dong Han Kim, Taehyun Yang, Youran Noh, Song Hee Hong

**Affiliations:** 1College of Pharmacy, Seoul National University, Seoul 08826, Republic of Korea; kimdh830@snu.ac.kr; 2Research Institute of Pharmaceutical Science, Seoul National University, Seoul 08826, Republic of Korea; 3Bulgwang Ildeung Pharmacy, Seoul 03396, Republic of Korea; yangth91@gmail.com; 4College of Pharmacy, Hanyang University, Ansan 15588, Republic of Korea

**Keywords:** patient medication reviews and ratings (PMRRs), Treatment Satisfaction Questionnaire for Medication (TSQM), hypertension, patient-reported outcomes, patient-centered care, online drug reviews

## Abstract

**Background/objective:** Online platforms for sharing prescription drug experiences are becoming increasingly available, yet their validity as measures of patient satisfaction remains unclear. This study aimed to evaluate the potential of an online drug review system, *WePharm*, as a proxy for treatment satisfaction among older adults taking antihypertensive medications. **Methods:** A cross-sectional survey using a convenience sample was conducted from February to July 2018 among patients aged 50–80 years recruited from four senior welfare centers and one community pharmacy in Seoul. Participants completed both an online review via *WePharm* and a paper-based Treatment Satisfaction Questionnaire for Medication (TSQM). Satisfaction attributes included drug efficacy, side effects, convenience, affordability, and willingness to recommend. Pearson correlation coefficients and ANOVA were used to examine concordance and associated factors. **Results:** A total of 313 participants were included. Online review scores were significantly correlated with TSQM scores across all domains as follows: effectiveness (r = 0.451), side effects (r = 0.363), convenience (r = 0.285), and overall satisfaction (r = 0.256), all *p* < 0.0001. Key factors associated with satisfaction included region, stage of hypertension, income, duration of antihypertensive use, and comorbidity count. **Conclusions:** Online patient medication reviews, as implemented in *WePharm*, demonstrated moderate correlation with validated treatment satisfaction measures. These findings support the potential utility of online drug review systems as complementary tools for capturing real-world patient experience and informing shared decision-making in clinical practice, and as these findings were from a convenience sample, further research is expected with the aim of improving generalizability.

## 1. Introduction

Patient medication reviews and ratings (PMRRs) on online platforms reflect patients’ real-world experiences with drug therapies. These narratives commonly address multiple facets of medication use, including effectiveness, safety, convenience, and affordability. As first-person accounts, they can engage readers and encourage patients to seek information, discuss options with providers, and participate in decisions—behaviors linked to better adherence and outcomes [1,2,3].

In the consumer markets, product reviews reduce uncertainty and guide choices [4,5,6]. Similarly, patient experience reviews (PERRs) help evaluate healthcare services and inform quality improvement [7,8,9]. While PMRRs do not determine prescriptions, they may shape preferences and foster dialogue with prescribers [10,11], and they can reveal adherence barriers or challenges with specific drugs [12,13,14,15,16,17,18].

Despite these potential benefits, questions remain regarding the validity and interpretability of PMRRs as a measure of treatment satisfaction [19,20]. One prior study suggested that patient ratings across six dimensions—effectiveness, adverse effects, cost, ease of use, food interactions, and overall satisfaction—are associated with adherence behavior and may reflect perceived treatment benefit [21]. Notably, four of these domains overlap with those in the Treatment Satisfaction Questionnaire for Medication (TSQM), a widely used and validated instrument. However, that study did not assess how PMRRs compare with TSQM scores, nor were the reviews collected in an actual online environment, limiting their generalizability to real-world contexts. Validating PMRRs is not only a methodological concern but also a practical one. Unlike conventional PRO instruments, PMRRs often capture patient-centered concerns—such as affordability, lifestyle compatibility, and real-world usability—that are underrepresented in standardized tools like TSQM. These dimensions are increasingly recognized as critical for shared decision-making, treatment adherence, and health equity. Moreover, PMRRs may offer scalable insights for pharmacovigilance and patient-focused drug development, especially when aggregated across large populations.

To address this gap, we developed *WePharm* (wepharm.snu.ac.kr), a digital platform for sharing patient medication experiences. This study aimed to assess the validity of PMRRs collected through *WePharm* as a proxy for TSQM scores among older adults using antihypertensive medications. Specifically, our objectives were the following: (1) to examine the correlation between PMRR and TSQM scores across total and subdomain measures, and (2) to evaluate the content validity of PMRRs in reflecting treatment satisfaction. By examining the strength and direction of these correlations, we sought to determine whether online PMRRs can provide a practical and efficient means of assessing treatment satisfaction in real-world clinical contexts.

## 2. Methods

### 2.1. Development of ‘WePharm’ Online Drug Reviews and Ratings Site

*WePharm* is an online ICT (Information and Communications Technology) system for drug experience sharing developed by the Social Pharmacy Laboratory of Seoul National University College of Pharmacy to allow patients and clinical experts to communicate about medications. It uses a website and a mobile app to collect voluntary patient experience data that can be freely shared amongst the public, patients, and experts. Through *WePharm*, patients will be able to learn about their drugs and conditions, which will likely lead to their active involvement in the decision-making process with their provider.

*WePharm* will include over-the-counter (OTC) drugs, as well as medications for hypertension, thyroid disorders, and pediatric diseases; as of now, only the antihypertensives are registered and categorized by generic and brand names. Users can search for medications by scrolling through the list or by using the search bar. The platform displays patient reviews (5-point scale), summary ratings by medication, and a top-10 list of the most reviewed drugs. Reviews rate the drug’s effects, side effects, convenience, affordability, interaction with food, and patient’s willingness to recommend the drug (Figure 1, demonstration of the portal page translated in English).

The main functions of *WePharm* are the following: “to do drug reviews” and “to view drug reviews”. The latter function displays the trade or brand name at the top. The reviews filtered into generic names are displayed under it. The user’s ID, age, sex, and the star rating the reviewers gave are viewable.

### 2.2. Study Design and Survey Administration

We conducted a cross-sectional survey of older adults taking antihypertensive medication, recruited by convenience sampling at senior welfare centers and a community pharmacy in Seoul, Republic of Korea. Eligible participants were 50–80 years old with a physician diagnosis of hypertension, were currently taking antihypertensive medication, and agreed to perform and complete the survey which was required for this study. Of these participants, (1) those diagnosed with hypertension but who have stopped taking antihypertensives in the past 6 months and (2) those who could not provide the name of the medication they were taking were excluded. The older adults were visitors of senior centers or a pharmacy in Seoul, Republic of Korea. The senior centers and the pharmacy were selected based on the willingness to collaborate in on-site data collection. Five senior centers and one pharmacy that agreed to participate were contacted to arrange on-site visits for survey administration and data collection.

According to sample size computation, the target sample size was determined to detect a moderate correlation (Pearson r ≈ 0.30) between PMRR and TSQM domains with 80% power, requiring at least 85 participants. Anticipating exclusions and incomplete responses, we aimed to recruit ≥500 individuals. Subgroup comparisons were exploratory and so interpreted descriptively.

Surveys were administered between February and July 2018 at five senior welfare centers (Seodaemun-gu, Guro-gu, Songpa-gu, Eunpyeong-gu, and Gangseo-gu) and one local pharmacy (Dobong-gu) in Seoul. A pilot survey was conducted with 7 visitors at the Seodaemun senior center to assess the feasibility and clarity of the questionnaire. Including these pilot participants, a total of 530 individuals completed the survey. After excluding 217 respondents due to duplicate entries from the pilot survey (*n* = 7) and ineligibility for study participation (*n* = 210), which was determined by incomplete survey responses (*n* = 182), age outside the eligible age of 50–80 years (*n* = 91), and no hypertension diagnosis (*n* = 19), the final analytic sample comprised 313 participants (Figure 2).

The survey followed a fixed sequence and instrument order was not randomized. After written informed consent, participants completed a paper questionnaire (including sociodemographic/clinical characteristics and the TSQM), took a brief rest, and then completed the online PMRR drug review on the *WePharm* platform. A brief rest period between the two instruments was given to minimize recall bias associated with completing both the PMRR and TSQM surveys. This interval was intended to allow for a short cognitive reset while avoiding changes in medication experience that could occur with a multi-day separation. During the PMRR survey, which was administered through the *WePharm* platform, some older participants experienced initial difficulties accessing the system. Therefore, researchers provided assistance with platform navigation (e.g., registration and locating the drug page) when needed. To minimize potential bias, all researchers were trained to avoid offering guidance or feedback related to the review content itself.

### 2.3. Ethical Consideration

This study was conducted in accordance with the ethical standards for human subject research. The study protocol (IRB No. 1705/003-007, Initial approval date: 22 May 2017) was reviewed and approved by the Institutional Review Board (IRB) of Seoul National University. All participants were provided with a clear explanation of the study’s purpose and procedures and gave written informed consent prior to participation. Participants were informed that all data collected would be used solely for research purposes and that confidentiality and anonymity would be strictly maintained. They were also notified of their right to withdraw from the survey at any time without penalty. Contact information was provided for any future inquiries. Upon completion of the survey, participants received a small token of appreciation for their participation.

### 2.4. Survey Instruments

#### 2.4.1. Sociodemographic and Health Characteristics

The general characteristics of the study participants were categorized based on the healthcare utilization model proposed by Andersen and Newman (1973) [22]. Sociodemographic variables included sex, age, marital status, education level, and income. Health status variables included the stage of hypertension, the number of antihypertensive medications currently taken, and the number of past or present comorbid conditions.

#### 2.4.2. PMRRs

Online surveys took approximately 10 min. Participants were registered to *WePharm* through either the website or mobile app, which collected information about the participant’s ID, sex, and date of birth. After registration, they answered questions about the name of the drug they use, the duration and reason for use, the level of satisfaction with the drug, and the effects and side effects of the drug. During the online review, the participants searched the antihypertensive medication they take and rated it based on “Satisfaction with the Drug”, which included the following 6 items: “effectiveness”, “side effects”, “ease of use”, “affordability”, “food interaction”, and “willingness to recommend (WTR)”. Rating was carried out on a Likert scale from 1 to 5, with 1 being “strongly disagree” and 5 being “strongly agree.” All construct measurements were carried out using single-item questions. While internal consistency cannot be computed for single-item measures, prior research supports their validity and utility in survey research (e.g., overall health status and satisfaction with care) [23,24].

#### 2.4.3. TSQM

Treatment satisfaction was measured using the Treatment Satisfaction Questionnaire for Medication (TSQM), developed by Atkinson et al. [25]. The Korean version of TSQM 1.4 was used in this study with permission from its translator [26]. The instrument comprises 14 items across the following four domains: Effectiveness, Side Effects, Convenience, and Global Satisfaction. For the Side Effects domain, respondents who had not experienced adverse effects were instructed to select “No” for the first item and skip the remaining items in that domain and were scored as 100 [27]. Each of the other domains contains three items.

Domain scores were transformed to a 0–100 scale, with higher scores indicating greater satisfaction [25]. The internal consistency (Cronbach’s alpha) of each domain, as reported by the original developers, was as follows: Effectiveness = 0.87, Side Effects = 0.84, Convenience = 0.86, and Global Satisfaction = 0.80.

The reliability of the instruments was assessed in the current sample. Internal consistency was evaluated using Cronbach’s alpha (α), with values above 0.70 considered acceptable. In this study, Cronbach’s α coefficients ranged from 0.78 to 0.91 across the scales, indicating good internal consistency.

### 2.5. Statistical Analysis

Analyses were conducted in SAS 9.4 statistical program and Excel 2016. Analyses were designed to evaluate the convergent validity between PMRR and TSQM domains. Descriptions of PMRRs and the TSQM by participant characteristics was carried out using one-way ANOVA (Analysis of Variance). If there were significant group differences, pair-wise comparisons by subgroups were also performed. Correlation between domains of PMRRs and the TSQM was carried out using Pearson’s r, and partial correlations adjusted for sex, age, education, comorbidity count, and polypharmacy. ANOVA and Pearson’s correlation were used to assess domain-level associations and known-group differences.

## 3. Results

### 3.1. Baseline Characteristics of the Study Population

During the study period, 530 participants were enrolled. A total of 217 participants were excluded; the reasons included survey incompletion (*n* = 182), age ineligibility as they were outside 50–80 years (*n* = 91), no hypertension diagnosis (*n* = 19), and pilot survey entries resulting into duplicates (*n* = 7). A total of 313 participants comprised the full analysis set (Figure 2). Among them, 199 (63.6%) were women and 114 (36.4%) were men (Table 1). The majority of participants were aged in their 70s (75.7%), followed by those in their 60s (15.3%) and 50s (9.0%). In terms of social and economic characteristics, 58.6% lived with a spouse. Educational attainment was relatively low overall, with nearly half (46.9%) completing only middle school or below, 29.5% finishing high school, and 23.6% attaining a college-level education or higher. More than half (51.3%) reported a monthly income of KRW 1 million or less, while 29.4% earned between KRW 1–3 million and 19.3% reported more than KRW 3 million.

Regarding blood pressure categories, more than half of the participants (52.1%) were classified as having Stage 1a hypertension (SBP 140–149 mmHg, DBP 90–99 mmHg), 19.2% upper Stage 1 hypertension (SBP 150–159 mmHg), and 13.7% Stage 2 hypertension (SBP ≥ 160 mmHg or DBP ≥ 100 mmHg). Most patients (84.4%) were on monotherapy, 12.8% used two medications, and 2.6% used three. Nearly half (47.0%) of the study’s participants reported a treatment duration of about 10 years.

The most common comorbidities were diabetes mellitus without complications (24.6%), connective tissue diseases or peptic ulcer disease (17.9%), peripheral vascular disease (16.0%), and cerebrovascular disease (12.5%). In terms of overall medication burden, 55.1% were taking one to three prescription drugs across all conditions. The average monthly copayment was modest, with the largest group (25.4%) reporting costs of KRW 10,000–20,000.

### 3.2. Description of Responses to PMRRs and the TSQM

The 313 study participants submitted a total of 390 completed reviews of antihypertensive drugs, covering 48 types. The drug class with the largest number of responses was combination antihypertensives (*n* = 15), followed by calcium channel blockers (*n* = 10), ACE inhibitors/ARBs (*n* = 9), and beta blockers (*n* = 6). Two reviews addressed combination products for hypertension and hyperlipidemia, and one was for the alpha blocker doxazosin.

Across PMRR domains, the highest score was observed for ‘food interaction’ (mean = 4.02, SD = 1.08), followed by ‘side effects’ (mean = 3.990, SD = 1.11) and ‘ease of use’ (mean = 3.987, SD = 1.04). The lowest score was for ‘willingness to recommend’ (mean = 3.25, SD = 1.32) (Table 2). For TSQM domains, the highest rating was in ‘side effects’ (mean = 95.98, SD = 12.50), followed by ‘convenience’ (mean = 65.23, SD = 15.26). ‘Effectiveness’ averaged 64.54 (SD = 17.12), and ‘global satisfaction’ was 62.62 (SD = 18.24). The particularly high TSQM side effect score reflects the fact that many participants reported no adverse events, which were coded at the maximum score of 100 [27] thereby elevating the overall mean, and participants who experienced side effects (*n* = 38) yielded an average of 66.89 (SD = 18.12) as still scored highest among TSQM domains.

### 3.3. Comparative Subgroup Analysis of PMRRs and the TSQM by Participant Characteristics

Patient satisfaction differed significantly across several socioeconomic and clinical subgroups, with consistent patterns observed across both PMRRs and the TSQM, though each instrument highlighted different dimensions (Table 3 and Table 4). Both PMRRs and the TSQM showed that ‘effectiveness’ ratings were strongly influenced by district, stage of hypertension, monthly income, and treatment duration. In PMRRs, Songpa/Dobong destricts gave the lowest scores (3.55, 3.80), while Guro district, the lowest income among all destricts, reported the highest score (4.19, *p* = 0.0056). The TSQM findings echoed these regional disparities, with Guro again showing the highest ratings (68.52, *p* = 0.0037) and Dobong/Songpa the lowest two (57.41, 62.83). Disease severity was also important, as follows: Stage 2 patients rated ‘effectiveness’ higher than Stage 1b patients in both PMRRs (4.26 vs. 3.57, *p* = 0.0016) and the TSQM (72.48 vs. 58.52, *p* = 0.0006). Higher income (KRW ≥ 3 million) and higher education (college or above) consistently predicted lower effectiveness in both measures (*p* < 0.01). Ratings improved with longer treatment duration, from < 1 year to > 10 years in both scales (*p* < 0.05).

In terms of ‘side effects’, PMRRs indicated that ‘side effects’ were well-tolerated with the highest domain rating of 4.0 across subgroups, except for different groups in income and treatment duration as follows: higher-income patients were less satisfied (*p* = 0.006), and patients in their first year of treatment gave lower ratings compared from over 10 years (3.29 vs. 4.08, *p* = 0.0436). However, the TSQM showed no meaningful variation across all subgroups (*p* > 0.1), suggesting broad tolerability across patients.

Both instruments in ‘ease of use/convenience’ domain highlighted district-level and treatment duration differences. PMRR ‘ease of use’ scores were lower in Songpa/Dobong (3.60/3.83) compared with Guro/Gangseo (4.25/4.06, *p* = 0.0123), and the TSQM showed lower ‘convenience’ in Seodaemun/Dobong (60.98/61.93) compared with Gangseo/Songpa (68.26/66.40, *p* = 0.0403). In both measures, Stage 2 patients rated ‘ease of use/convenience’ higher than Stage 1b patients, and scores increased with longer treatment duration (*p* < 0.05). PMRRs additionally revealed the influence of income and monthly out-of-pocket (OOP) drug costs, with higher values linked to lower ease of use (*p* = 0.0063, *p* = 0.0060).

PMRRs explicitly measured ‘affordability’, showing that patients with more comorbidities rated drug therapy affordability higher (*p* = 0.0221), while those with higher monthly OOP drug costs (≥30,000 KRW) rated it lower (*p* < 0.0001). The TSQM did not include a direct affordability scale but reflected similar patterns through effectiveness and global satisfaction, both of which declined with higher OOP drug costs (*p* < 0.05). Only PMRRs captured ‘food interaction’ domain. Patients with lower income (*p* = 0.0185) and longer treatment duration (*p* = 0.0077) reported fewer problems with food–drug interaction.

PMRRs measured global satisfaction using the wording of ‘willingness to recommend’ while the TSQM used three questionnaire items. Both instruments demonstrated similar subgroup effects. Willingness to recommend was lowest in Songpa (2.79) and higher in Guro (3.52, *p* = 0.0124), with lower ratings among college-educated (2.92, *p* = 0.033) and higher ratings among married patients (3.52 vs. 3.08, *p* = 0.0037). TSQM global satisfaction results aligned similarly although statistical significance was found differently. Patients with Stage 2 hypertension, lower education, and lower income reported significantly higher satisfaction (*p* < 0.05). Longer treatment duration was significantly associated with higher global satisfaction (66.70 for ≥ 10 years vs. 53.27 for <1 year, *p* = 0.0015) in the TSQM but failed to reach statistical significance in PMRRs.

Overall, subgroup patterns were broadly consistent between PMRRs and the TSQM, with both measures showing lower satisfaction among higher-income and more educated patients, and higher satisfaction with longer treatment duration. Regional disparities also emerged in both instruments, with the poorest district reporting the highest satisfaction. Differences seemed to lie mainly in domain sensitivity, as follows: PMRRs uniquely captured affordability and food interaction, where TSQM has no measurement. PMRRs also captured comorbidity effects, while TSQM emphasized gender and age differences and provided more stable side effect ratings. Together, the findings underscore the importance of socioeconomic context, disease severity, and treatment experience in shaping patient satisfaction with antihypertensive therapy.

### 3.4. Correlation Between PMRR and TSQM Domains

PMRR domains were significantly correlated with the corresponding TSQM dimensions, though the strength of associations varied (Table 5). Effectiveness in PMRRs was most strongly correlated with TSQM Effectiveness (r = 0.451, *p* < 0.0001), followed by Global Satisfaction (r = 0.384, *p* < 0.0001). Smaller but significant correlations were also observed with TSQM Convenience (r = 0.240, *p* < 0.0001) and Side Effects (r = 0.145, *p* < 0.05). Side Effects in PMRRs aligned closely with TSQM Side Effects (r = 0.364, *p* < 0.0001) and was also correlated with Effectiveness (r = 0.400, *p* < 0.0001) and Global Satisfaction (r = 0.354, *p* < 0.0001). A more modest association was found with Convenience (r = 0.133, *p* = 0.0183). Ease of using medication in PMRRs correlated with TSQM Convenience (r = 0.286, *p* < 0.0001), although its associations were actually stronger with TSQM Effectiveness (r = 0.318, *p* < 0.0001) and Global Satisfaction (r = 0.321, *p* < 0.0001). Willingness to recommend in PMRRs was modestly associated with all four TSQM domains, the strongest being with Global Satisfaction (r = 0.256, *p* < 0.0001) and Effectiveness (r = 0.271, *p* < 0.0001) and being weaker with Side Effects (r = 0.193, *p* = 0.0006) and Convenience (r = 0.120, *p* = 0.0345). A partial correlation model with adjusting covariates (sex, age, education, comorbidity, and polypharmacy) has also been performed and showed similar results.

Food interaction and affordability domains in PMRRs have no counterpart domains in the TSQM. Affordability demonstrated significant but generally weaker correlations across TSQM domains, yielding the highest value for Effectiveness (r = 0.233, *p* < 0.0001) and Global Satisfaction (r = 0.222, *p* < 0.0001). Food interaction correlated moderately with TSQM Global Satisfaction (r = 0.324, *p* < 0.0001) and Effectiveness (r = 0.307, *p* < 0.0001), while weaker associations were observed with Side Effects and Convenience.

## 4. Discussions

This study sought to test the validity of patient medication reviews and ratings (PMRRs) against the established Treatment Satisfaction Questionnaire for Medication (TSQM) in patients taking antihypertensive drugs. While PMRR data were collected online through *WePharm*, TSQM data were obtained through paper surveys, allowing for comparison between digitally crowdsourced reviews and validated self-reported treatment satisfaction.

### 4.1. Key Study Findings

PMRR scores varied significantly by demographic and socioeconomic characteristics such as district, stage of hypertension, education, income, duration of antihypertensive use, and monthly copayment. In contrast, TSQM scores were additionally influenced by sex and age, suggesting that standardized measures may be more sensitive to demographic factors than online reviews. Notably, PMRRs ranked highest in the domains of ease of use and effectiveness, while the TSQM scored highest in effectiveness and global satisfaction. When interpreted together, a consistent trend emerged as follows: higher scores were reported by older female patients with lower copayments, while lower scores were given by younger, more affluent, and more educated patients with fewer comorbidities. This pattern is consistent with prior research indicating that women may report greater satisfaction with healthcare experiences, possibly due to differences in expectations, communication preferences, or engagement with treatment. A large U.S. study demonstrated that older age and female sex were independently associated with higher patient satisfaction across healthcare settings [28]. Similarly, previous systematic research reported that age, sex, marital status, and socioeconomic status significantly influence satisfaction, with older patients and women consistently reporting higher satisfaction [29]. These findings suggest that gender may play a role in shaping perceived treatment benefit and satisfaction, and they highlight the importance of considering sociodemographic factors in the interpretation of patient-reported outcomes. Further research is needed to explore the underlying mechanisms and cultural influences that may contribute to these differences.

The relationship of satisfaction with socioeconomic status and education is more complex. Patients with lower income reported higher satisfaction when copayments were low, which is consistent with prior evidence showing that cost burden is a strong determinant of satisfaction [17,18]. In contrast, more educated and affluent patients tended to give lower satisfaction scores, possibly due to higher expectations regarding treatment effectiveness. This interpretation is supported by studies showing that individuals with higher educational attainment and resources often set higher standards for their care, leading to more critical evaluations despite greater access to services [30]. Taking together, these results suggest that patient satisfaction is influenced not only by the quality but also by expectations shaped by demographic and socioeconomic contexts.

### 4.2. Convergence Between PMRRs and the TSQM

PMRR domains showed moderate correlations with the corresponding TSQM constructs. Specifically, PMRR ‘effectiveness’, ‘side effects’, and ‘ease of use’ were strongly associated with TSQM ‘effectiveness’, ‘side effects’, and ‘convenience’, respectively. PMRR ‘willingness to recommend’ aligned most closely with TSQM ‘global satisfaction’. Importantly, PMRRs also introduced unique domains—‘affordability’ and ‘food interaction’—not captured by TSQM. As expected, these domains showed weaker correlations due to the lack of conceptual overlap.

Given that PMRR domains are single-item constructs, internal consistency metrics and confirmatory factor analysis are not applicable. Therefore, our validity assessment focused on convergent patterns with TSQM domains and known-group differences, providing preliminary evidence of construct alignment rather than structural equivalence. Nevertheless, PMRR domains showed significant correlations with treatment satisfaction, suggesting moderate convergent validity while expanding the scope to include patient concerns highly relevant in real-world practice, such as cost and dietary compatibility.

Although no prior studies have directly compared PMRRs with the TSQM, the TSQM has been widely validated across conditions [19,25,27,31]. Prior work has also shown consistent demographic influences on satisfaction as follows: older age and female sex are generally associated with higher satisfaction, whereas higher education and income are often linked to lower satisfaction, likely due to elevated expectations [28,29]. Although this study did not include qualitative data, future mixed-methods research incorporating patient interviews or narrative analyses of PMRR content could further elucidate how expectations shape satisfaction across socioeconomic groups.

The moderate correlations observed between PMRR and TSQM domains (r ≈ 0.35–0.58) are consistent with those reported among other validated patient satisfaction or treatment experience instruments in chronic disease settings. Prior studies evaluating concordance between instruments such as the Treatment Satisfaction Questionnaire for Medication (TSQM), the Diabetes Treatment Satisfaction Questionnaire (DTSQ), and the Patient Satisfaction with Medication Questionnaire (PSMQ) have typically shown correlation coefficients in the range of 0.3–0.6, reflecting moderate construct convergence rather than redundancy [19,25]. This consistency supports the convergent validity of PMRRs as a complementary measure of treatment satisfaction in real-world populations.

### 4.3. Implications for Practice and Policy

The results indicate that online review platforms such as *WePharm* can generate meaningful and valid measures of treatment satisfaction, showing moderate convergent validity with traditional survey instruments. By incorporating dimensions such as affordability and food interaction, PMRRs provide a broader perspective on the patient experience. These insights may support shared decision-making between patients and healthcare providers and help identify barriers to adherence in real-world settings.

From a policy perspective, platforms like *WePharm* could contribute to patient-centered care by surfacing real-world concerns—such as cost burden and regimen usability—that are often underrepresented in conventional PROs. While PMRR data may complement clinical trial evidence and post-marketing surveillance, their use in reimbursement decisions or clinical practice guidelines would require further empirical validation, standardization, and governance.

### 4.4. Limitations and Future Directions

This study has several limitations. First, the convenience sample of participants with —primarily older adults recruited from senior centers and one community pharmacy located in Seoul, South Korea—limits the generalizability of the findings. The use of convenience sampling may introduce selection bias, and the geographical restriction to a single metropolitan area further constrains the external validity of the results in representing the broader Korean population. Also, many participants required assistance with digital platforms, which implies the digital literacy of participants may have influenced the quality of PMRR data, introducing interviewer assistance-related biases, although the assistance was limited to navigation. Furthermore, excluded participants with survey incompletion tended to be older and have lower educational attainment [30]. Future studies should involve a broader geographical area, with diverse age range and digital literacy profiles. District-level variation in satisfaction ratings was observed, likely reflecting differences in socioeconomic conditions and healthcare resources across regions. However, because data were collected from a limited number of conveniently selected districts, the findings may not be generalized to all districts within Seoul.

Another limitation lies in the difference in survey administration modes between the online PMRRs and the paper-based TSQM. Although both instruments were completed sequentially during the same session under consistent conditions, subtle mode-related differences in response behavior cannot be entirely ruled out. A related limitation arises from the sequential administration of PMRRs and the TSQM, which may have introduced memory or carryover effects as participants completed both surveys during the same session. Because PMRRs and the TSQM were completed in one session, carryover or priming effects are possible despite the rest interval. Because instrument order was not randomized and assistance/timing were not tracked, we were unable to quantify these effects. Future studies will counterbalance order, evaluate mode equivalence, and record assistance, device type, and timestamps to assess potential bias. Furthermore, because PMRR domains are single-item measures, we did not perform formal psychometric validation (e.g., test–retest reliability or cognitive interviewing) and future studies will undertake these procedures to further establish content validity.

Specific to the Korean medication dispensing system, the multi-dose pouch dispensing practice posed challenges for accurate patients’ self-identification of individual medication, particularly among those taking multiple drugs. Although the medication information sheet provided by pharmacists included medication-level details, the expanded use of ICT-linked codes—allowing patients to access digital drug information directly from their dispensing sheet—will be needed to reduce patient-level misclassification and enable more precise medication attribution.

This study focused exclusively on antihypertensive medications to ensure feasibility and internal consistency during the initial validation of the *WePharm* platform. Consequently, the generalizability of the findings to other therapeutic classes may be limited, as patient engagement and satisfaction with online medication reviews could vary by disease type, treatment complexity, and perceived medication risk. Future research should examine whether the observed validity patterns persist across other drug classes as *WePharm* expands to include medications for chronic conditions such as diabetes, dyslipidemia, and thyroid disorders. Comparative analyses across therapeutic areas could clarify whether disease-specific factors influence the correlation between online reviews and standardized satisfaction measures.

## 5. Conclusions

This study demonstrates that PMRRs collected through an online platform can capture treatment satisfaction in a way that is both convergent with the validated TSQM and more reflective of patient-centered concerns such as cost and food interactions. *WePharm* thus has the potential to become a valuable resource for patients, providers, and policymakers. By bridging the gap between provider-led care and patient-involved care, PMRR platforms can enhance treatment satisfaction, support adherence, and ultimately improve the quality of chronic disease management in Korea.

## Figures and Tables

**Figure 1 healthcare-13-02918-f001:**
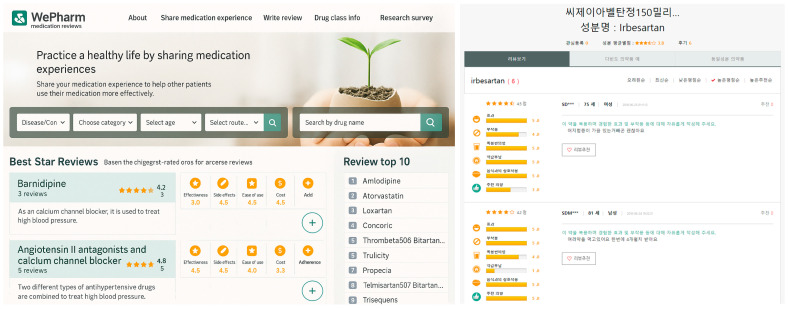
Demonstration of *WePharm* online drug review system.

**Figure 2 healthcare-13-02918-f002:**
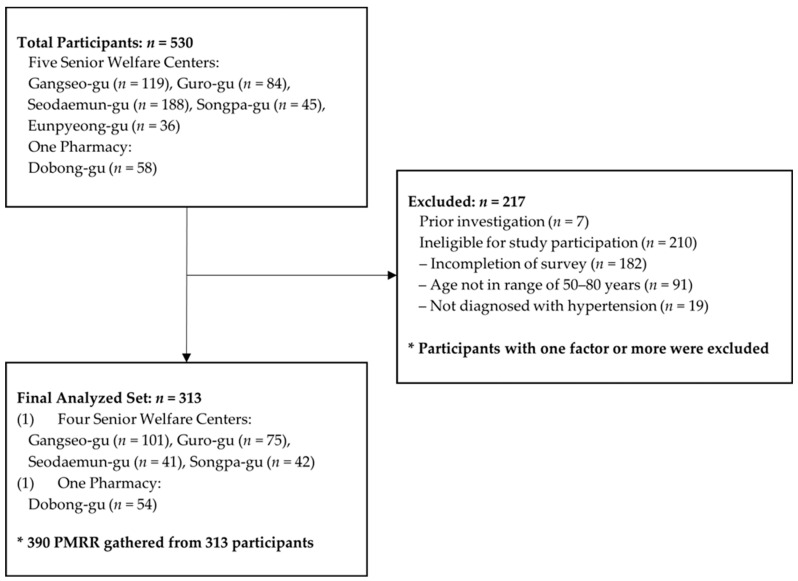
Flowchart of disposition of study participants.

**Table 1 healthcare-13-02918-t001:** Sociodemographic and hypertension characteristics.

Participant Characteristics	N (%)
**Sex**	
Male	114 (36.42)
Female	199 (63.58)
**Age (Years)**	
50s	28 (8.95)
60s	48 (15.34)
70s	237 (75.72)
**District**	
Seodaemun	41 (13.10)
Gangseo	101 (32.27)
Songpa	42 (13.42)
Guro	75 (23.96)
Dobong	54 (17.25)
**Spouse**	
Yes	181 (58.58)
No	128 (41.42)
**Education**	
≤Middle school graduate	145 (46.93)
High school graduate	91 (29.45)
≥College graduate	73 (23.62)
**Monthly Income (KRW)**	
≤1 million	136 (51.32)
>1 million, ≤3 million	78 (29.43)
>3 million	51 (19.25)
**Comorbidity ***	
DM without complications	77 (24.60)
Connective tissue disease	56 (17.89)
Digestive ulcer	56 (17.89)
Peripheral vascular disease	50 (15.97)
Cerebrovascular disease	39 (12.46)
Others	114 (36.42)
**Stage of Hypertension (Systolic/Diastolic, mmHg)**	
Not reported	37 (11.82)
Stage 1a (140–149/90–99)	163 (52.08)
Stage 1b (150–159/90–99)	60 (19.17)
Stage 2 (≥160/≥100)	43 (13.74)
Isolated Systolic (≥140/<90)	10 (3.19)
**No. of Antihypertensives**	
1	264 (84.35)
2	40 (12.78)
3	8 (2.56)
4	1 (0.32)
**Duration of Antihypertensive Use (Years)**	
<1 year	17 (5.43)
≥1, <3 years	35 (11.18)
≥3, <5 years	54 (17.25)
≥5, <10 years	60 (19.17)
>10 years	147 (46.96)
**No. of Prescription Drugs Including Antihypertensives**	
1–3	172 (55.1)
4–6	92 (29.49)
7–9	30 (9.62)
≥10	16 (5.13)
Don’t Know	2 (0.64)
**Monthly Drug OOP Cost (KRW)**	
<5000	73 (23.47)
≥5000, <10,000	47 (15.11)
≥10,000, <20,000	79 (25.40)
≥20,000, <30,000	48 (15.43)
≥30,000	64 (20.58)

Note: * One person can have multiple comorbid diseases, and past or present comorbidities were both included; KRW: Korean Won; OOP: Out of Pocket.

**Table 2 healthcare-13-02918-t002:** Comparison of responses to PMRRs and the TSQM.

Domains	PMRRs (*n* = 390)		TSQM (*n* = 313)	
	Mean ± SD	Median, Range	Mean ± SD	Median, Range
Effectiveness	3.95 ± 0.95	4, 1–5	64.54 ± 17.12	66.67, 5.56–100
Side effects	3.99 ± 1.11	4, 1–5	95.98 ± 12.50	100, 33.33–100
Ease of Use/Convenience	3.99 ± 1.04	4, 1–5	65.23 ± 15.26	66.67, 22.22–100
Affordability	3.55 ± 1.27	3, 1–5	NA	NA
Food interaction	4.02 ± 1.08	4, 1–5	NA	NA
WTR/Global Satisfaction	3.25 ± 1.32	3, 1–5	62.62 ± 18.24	58.33, 11.11–100

Note: WTR: willingness to recommend; NA: not available.

**Table 3 healthcare-13-02918-t003:** PMRRs by participant characteristics.

Characteristics	N	Effectiveness	Side Effects	Ease of Use	Affordability	Food Interaction	WTR
		Mean ± SD (95% CI)	Mean ± SD (95% CI)	Mean ± SD (95% CI)	Mean ± SD (95% CI)	Mean ± SD (95% CI)	Mean ± SD (95% CI)
**Sex (Prob)**		0.1231	0.9922	0.8621	0.7617	0.2868	0.6721
Male	114	3.84 ± 0.94 (3.67–4.02) 4.02 ± 0.96 (3.88–4.15)	3.99 ± 1.07 (3.79–4.19) 3.99 ± 1.14 (3.83–4.15)	3.97 ± 0.96 (3.80–4.15) 3.99 ± 1.08 (3.84–4.15)	3.52 ± 1.17 (3.30–3.73) 3.56 ± 1.32 (3.38–3.75)	3.93 ± 1.09 (3.73–4.13) 4.07 ± 1.07 (3.92–4.21)	3.21 ± 1.20 (2.99–3.43) 3.28 ± 1.39 (3.08–3.47)
Female	199						
**Age, Years (Prob)**		0.3828	0.4877	0.1775	0.5199	0.1601	0.7819
50s	28	3.71 ± 0.89 (3.36–4.07) 3.96 ± 0.94 (3.69–4.23) 3.98 ± 0.96 (3.86–4.10)	3.75 ± 1.14 (3.34–4.16) 4.00 ± 1.07 (3.68–4.32) 4.02 ± 1.12 (3.87–4.16)	3.64 ± 0.98 (3.26–4.03) 3.98 ± 1.02 (3.68–4.27) 4.03 ± 1.04 (3.90–4.16)	3.32 ± 1.24 (2.85–3.79) 3.67 ± 1.13 (3.31–4.03) 3.55 ± 1.29 (3.39–3.71)	3.64 ± 1.09 (3.24–4.04) 4.04 ± 1.03 (3.74–4.35) 4.05 ± 1.08 (3.92–4.19)	3.25 ± 0.96 (2.76–3.74) 3.38 ± 1.29 (3.00–3.75) 3.23 ± 1.36 (3.06–3.40)
60s	48						
70s	237						
**District (Prob)**		0.0056	0.2082	0.0123	0.3896	0.4778	0.0124
Songpa	42	3.55 ± 0.99 (3.26–3.83)	3.76 ± 1.34 (3.42–4.10)	3.60 ± 1.14 (3.28–3.91)	3.31 ± 1.29 (2.92–3.69)	3.90 ± 1.18 (3.58–4.23)	2.79 ± 1.33 (2.39–3.18)
Gangseo	101	4.04 ± 0.92 * (3.86–4.22)	3.96 ± 1.07 (3.74–4.18)	4.06 ± 1.01 (3.84–4.26)	3.49 ± 1.33 (3.24–3.73)	3.99 ± 1.14 (3.78–4.20)	3.06 ± 1.47 (2.80–3.31)
Seodaemun	41	3.93 ± 0.90 (3.64–4.22)	4.02 ± 0.98 (3.68–4.37)	3.93 ± 0.98 (3.61–4.24)	3.56 ± 1.22 (3.17–3.95)	3.93 ± 1.01 (3.59–4.26)	3.46 ± 1.02 (3.06–3.86)
Guro	75	4.19 ± 0.95 * (3.97–4.40)	4.23 ± 1.07 (3.97–4.48)	4.25 ± 1.00 * (4.02–4.49)	3.77 ± 1.27 (3.49–4.06)	4.21 ± 1.01 (3.97–4.46)	3.52 ± 1.33 * (3.22–3.82)
Dobong	54	3.80 ± 0.91 (3.55–4.05)	3.87 ± 1.11 (3.57–4.17)	3.83 ± 1.00 (3.56–4.11)	3.52 ± 1.12 (3.18–3.86)	3.94 ± 1.01 (3.65–4.23)	3.44 ± 1.04 (3.10–3.79)
**Stage of Hypertension (Prob)**		0.0016	0.275	0.0381	0.0991	0.5116	0.1822
Stage 1a	163	4.02 ± 0.93 (3.88–4.17)	4.01 ± 1.14 (3.84–4.18)	4.12 ± 1.00 (3.97–4.28)	3.69 ± 1.26 (3.50–3.89)	4.07 ± 1.05 (3.90–4.23)	3.33 ± 1.32 (3.13–3.53)
Stage 1b	60	3.57 ± 0.94 * (3.33–3.80)	3.80 ± 1.08 (3.52–4.08)	3.68 ± 1.06 * (3.42–3.94)	3.22 ± 1.15 (2.90–3.54)	3.85 ± 1.08 (3.58–4.12)	3.05 ± 1.11 (2.72–3.38)
Stage 2	43	4.26 ± 0.95 ^†^ (3.97–4.54)	4.23 ± 1.06 (3.90–4.57)	4.09 ± 1.04 (3.79–4.40)	3.53 ± 1.42 (3.16–3.91)	4.12 ± 1.13 (3.80–4.44)	3.37 ± 1.41 (2.98–3.76)
Isolated systolic	10	4.00 ± 0.81 (3.42–4.58)	4.10 ± 0.87 (3.41–4.79)	3.90 ± 0.87 (3.26–4.54)	3.60 ± 0.96 (2.82–4.38)	3.90 ± 0.87 (3.23–4.57)	2.60 ± 1.64 (1.79–3.41)
**Spouse (Prob)**		0.1157	0.3273	0.6206	0.1254	0.9213	0.0037
Yes	181	3.87 ± 0.93 (3.73–4.01)	3.93 ± 1.12 (3.76–4.09)	3.96 ± 1.01 (3.80–4.11)	3.46 ± 1.24 (3.28–3.65)	4.01 ± 1.05 (3.85–4.17)	3.08 ± 1.27 (2.89–3.27)
No	128	4.05 ± 0.97 (3.88–4.21)	4.05 ± 1.10 (3.86–4.25)	4.02 ± 1.08 (3.83–4.20)	3.69 ± 1.28 (3.47–3.91)	4.02 ± 1.12 (3.83–4.21)	3.52 ± 1.34 (3.30–3.75)
**Education (Prob)**		0.0523	0.1080	0.3523	0.3096	0.4661	0.0330
≤Middle school	145	3.99 ± 1.01 (3.84–4.15)	4.06 ± 1.11 (3.87–4.24)	3.97 ± 1.09 (3.80–4.14)	3.58 ± 1.31 (3.37–3.79)	4.03 ± 1.10 (3.85–4.20)	3.41 ± 1.33 (3.19–3.62)
High school	91	4.05 ± 0.91 (3.86–4.25)	4.05 ± 1.13 (3.83–4.28)	4.10 ± 1.05 (3.88–4.31)	3.64 ± 1.32 (3.39–3.91)	4.10 ± 1.12 (3.88–4.32)	3.30 ± 1.36 (3.03–3.57)
≥College	73	3.71 ± 0.85 (3.49–3.93)	3.74 ± 1.06 (3.48–4.00)	3.86 ± 0.91 (3.62–4.10)	3.36 ± 1.07 (3.07–3.65)	3.89 ± 0.98 (3.64–4.14)	2.92 ± 1.18 * (2.62–3.22)
**Monthly Income (Prob)**		0.0096	0.0060	0.0063	0.0871	0.0185	0.0651
≤1M KRW	136	4.04 ± 0.92 (3.89–4.20)	4.13 ± 1.02 (3.95–4.30)	4.09 ± 0.99 (3.92–4.25)	3.58 ± 1.28 (3.37–3.79)	4.09 ± 1.07 (3.91–4.26)	3.31 ± 1.33 (3.09–3.53)
>1M, ≤3M	78	4.05 ± 0.89 (3.85–4.26)	4.04 ± 1.03 (3.80–4.28)	4.15 ± 0.94 (3.94–4.38)	3.72 ± 1.19 (3.45–3.99)	4.09 ± 0.94 (3.86–4.32)	3.37 ± 1.32 (3.08–3.66)
≥3M	51	3.61 ± 0.91 *^,†^ (3.36–3.86)	3.57 ± 1.17 *^,†^ (3.28–3.86)	3.63 ± 0.99 *^,†^ (3.36–3.90)	3.24 ± 1.10 (2.90–3.57)	3.63 ± 1.09 *^,†^ (3.34–3.91)	2.86 ± 1.14 (2.51–3.22)
**Comorbidity (Prob)**		0.2967	0.1789	0.7477	0.0221	0.4620	0.0012
0	221	3.90 ± 0.94 (3.77–4.03)	3.92 ± 1.14 (3.78–4.07)	3.96 ± 1.02 (3.82–4.10)	3.46 ± 1.25 (3.30–3.63)	4.00 ± 1.08 (3.86–4.14)	3.10 ± 1.30 (2.93–3.27)
1	55	4.04 ± 0.88 (3.78–4.29)	4.07 ± 1.01 (3.78–4.37)	4.07 ± 1.03 (3.80–4.35)	3.53 ± 1.34 (3.19–3.86)	3.95 ± 1.00 (3.66–4.23)	3.42 ± 1.28 (3.07–3.76)
≥2	37	4.14 ± 1.08 (3.83–4.44)	4.27 ± 1.04 (3.91–4.63)	4.03 ± 1.14 (3.69–4.36)	4.08 ± 1.11 (3.67–4.49)	4.22 ± 1.15 (3.87–4.57)	3.92 ± 1.25 (3.50–4.34)
**Treatment duration** **(Prob)**		0.0325	0.0436	0.0327	0.5226	0.0077	0.6186
<1 year	17	3.47 ± 0.94 (3.02–3.92)	3.29 ± 1.16 (2.77–3.82)	3.35 ± 0.99 (2.86–3.84)	3.29 ± 0.98 (2.69–3.90)	3.47 ± 1.06 (2.96–3.98)	3.00 ± 1.00 (2.37–3.63)
≥1, <3 years	35	3.68 ± 0.96 (3.37–4.00)	3.80 ± 1.07 (3.43–4.17)	3.74 ± 0.91 (3.40–4.09)	3.57 ± 1.22 (3.15–3.99)	3.77 ± 1.00 (3.42–4.13)	3.23 ± 1.26 (2.79–3.67)
≥3, <5 years	54	3.85 ± 0.92 (3.60–4.10)	3.93 ± 1.13 (3.63–4.22)	3.94 ± 1.03 (3.67–4.22)	3.33 ± 1.21 (2.99–3.67)	3.81 ± 1.16 (3.53–4.10)	3.06 ± 1.33 (2.70–3.41)
≥5, <10 years	60	4.05 ± 0.89 (3.81–4.29)	4.12 ± 1.04 (3.84–4.40)	4.08 ± 1.01 (3.82–4.34)	3.70 ± 1.29 (3.38–4.02)	4.33 ± 0.91 * (4.06–4.60)	3.38 ± 1.29 (3.05–3.72)
≥10 years	147	4.07 ± 0.97 (3.91–4.22)	4.08 ± 1.11 * (3.91–4.27)	4.10 ± 1.06 * (3.93–4.26)	3.59 ± 1.31 (3.38–3.79)	4.08 ± 1.09 (3.91–4.25)	3.31 ± 1.37 (3.09–3.52)
**Monthly** **Drug OOP** **(Prob)**		0.0688	0.2933	0.0060	<0.0001	0.2549	0.0815
<5K KRW	73	4.15 ± 0.98 (3.93–4.37)	4.16 ± 1.05 (3.91–4.42)	4.14 ± 1.01 (3.90–4.37)	4.21 ± 1.06 (3.93–4.48)	4.15 ± 1.11 (3.90–4.40)	3.62 ± 1.35 (3.31–3.92)
≥5K, <10K	47	3.83 ± 0.91 (3.56–4.10)	3.85 ± 1.21 (3.53–4.17)	3.85 ± 1.02 (3.58–4.14)	3.53 ± 1.15 * (3.19–3.88)	4.02 ± 1.11 (3.71–4.33)	3.06 ± 1.30 (2.69–3.44)
≥10K, <20K	79	4.05 ± 0.86 (3.84–4.26)	4.00 ± 1.06 (3.75–4.25)	4.20 ± 0.89 (3.98–4.43)	3.53 ± 1.25 * (3.27–3.80)	4.06 ± 1.01 (3.82–4.30)	3.22 ± 1.31 (2.92–3.51)
≥20K, <30K	48	3.92 ± 0.79 (3.65–4.19)	4.10 ± 0.97 (3.79–4.42)	4.04 ± 0.92 (3.75–4.33)	3.42 ± 1.18 * (3.08–3.76)	4.08 ± 0.87 (3.78–4.39)	3.02 ± 1.32 (2.65–3.39)
≥30K	64	3.72 ± 1.11 (3.49–3.95)	3.80 ± 1.25 (3.52–4.07)	3.61 ± 1.22 *^,‡^ (3.36–3.86)	2.91 ± 1.30 *^,‡^ (2.61–3.20)	3.75 ± 1.22 (3.48–4.02)	3.17 ± 1.25 (2.85–3.49)

Note: * Significant difference compared with the first subgroup (*p* < 0.05); ^†^ significant difference compared with the second subgroup (*p* < 0.05); and ^‡^ significant difference compared with the third subgroup (*p* < 0.05). CI: confidence interval; WTR: willingness to recommend; OOP: Out of Pocket; and KRW: Korean Won.

**Table 4 healthcare-13-02918-t004:** TSQM by participant characteristics.

Characteristics	N	Effectiveness	Side Effects	Convenience	Global Satisfaction
		Mean ± SD (95% CI)	Mean ± SD (95% CI)	Mean ± SD (95% CI)	Mean ± SD (95% CI)
**Sex** **(Prob)**		0.0005	0.6394	0.8477	0.0219
Male	114	60.09 ± 13.52 (57.61–62.57)	96.42 ± 11.20 (94.36–98.47)	65.45 ± 15.30 (62.64–68.26)	59.50 ± 16.36 (56.50–62.51)
Female	199	67.09 ± 18.43 (64.53–69.65)	95.73 ± 13.21 (93.89–97.56)	65.10 ± 15.27 (62.98–67.22)	64.41 ± 19.04 (61.76–67.05)
**Age, Year (Prob)**		0.0088	0.7293	0.7160	0.1368
50s	28	57.34, 11.32 (51.05–64.64)	97.67, 8.44 (92.96–102.28)	63.10, 15.80 (57.41–68.78)	56.65, 16.36 (49.89–63.41)
60s	48	60.76, 15.47 (55.96–65.57)	96.35, 11.52 (92.80–99.91)	65.97, 13.09 (61.63–70.32)	61.23, 16.93 (56.06–66.39)
70s	237	66.15, 17.70 * (63.99–68.31)	95.71, 13.10 (94.11–97.31)	65.33, 15.63 (63.38–67.29)	63.61, 18.61 (61.28–65.93)
**District**		0.0037	0.2829	0.0403	0.1428
Songpa	42	62.83 ± 15.34 (57.73–67.93)	92.46 ± 18.01 (88.67–96.25)	66.40 ± 12.69 (61.81–70.99)	61.97 ± 18.57 (56.46–67.48)
Gangseo	101	66.50 ± 16.71 (63.21–69.79)	96.37 ± 11.26 (93.93–98.81)	68.26 ± 14.04 (65.30–71.22)	64.91 ± 18.43 (61.35–68.46)
Seodaemun	41	63.55 ± 20.64 (58.39–68.71)	95.93 ± 12.23 (92.10–99.77)	60.98 ± 20.31 (56.33–65.62)	62.53 ± 16.95 (56.96–6.11)
Guro	75	68.52 ± 18.59 (64.70–72.34)	97.78 ± 9.52 (94.94–100.61)	65.19 ± 14.33 (61.75–68.62)	63.89 ± 19.07 (59.76–68.01)
Dobong	54	57.41 ± 11.26 ^†,※^ (52.91–61.91)	95.52 ± 13.22 (92.18–98.87)	61.93 ± 15.24 (57.89–65.98)	57.15 ± 16.78 (52.29–62.01)
**Stage of** **Hypertension** **(Prob)**		0.0006	0.7708	0.1685	0.0183
Stage 1a	163	65.06 ± 16.50 (62.47–67.66)	96.88 ± 9.84 (95.10–98.66)	66.77 ± 15.38 (64.39–69.14)	62.37 ± 17.78 (59.58–65.16)
Stage 1b	60	58.52 ± 15.29 (54.24–62.80)	96.11 ± 12.41 (93.17–99.05)	62.04 ± 16.27 (58.12–65.95)	57.69 ± 18.22 (53.09–62.28)
Stage 2	43	72.48 ± 20.82 ^‡^ (67.43–77.53)	94.96 ± 14.90 (91.49–98.43)	67.18 ± 13.91 (62.56–71.81)	69.12 ± 19.30 (63.69–74.55)
Isolated systolic	10	60.56 ± 9.95 (50.08–71.03)	95.00 ± 15.81 (87.80–102.20)	62.22 ± 16.31 (52.63–71.81)	60.00 ± 16.36 (48.75–71.25)
**Spouse (Prob)**		0.0566	0.1348	0.5279	0.1372
Yes	181	62.86 ± 16.46 (60.37–65.35)	95.03 ± 13.45 (93.19–96.86)	64.76 ± 14.90 (62.52–67.01)	61.43 ± 18.03 (58.78–64.09)
No	128	66.62 ± 17.79 (63.66–69.58)	97.20 ± 11.13 (95.02–99.38)	65.89 ± 16.01 (63.21–68.56)	64.56 ± 18.37 (61.40–67.72)
**Education (Prob)**		0.0014	0.3795	0.5545	0.0011
≤Middle school	145	66.48 ± 18.51 (63.74–69.21)	95.92 ± 13.07 (93.86–97.97)	64.94 ± 15.58 (62.44–67.45)	64.69 ± 18.29 (61.76–67.62)
High school	91	65.93 ± 16.35 (62.48–69.39)	97.16 ± 10.33 (94.57–99.76)	66.48 ± 16.25 (63.32–69.64)	64.80 ± 18.40 (61.10–68.50)
≥College	73	58.07 ± 13.06 *^,‡^ (54.21–61.92)	94.40 ± 14.03 (91.51–97.30)	63.93 ± 13.45 (60.40–67.45)	55.75 ± 16.57 *^,‡^ (51.61–59.88)
**Monthly Income (Prob)**		0.0003	0.1742	0.3046	0.0011
≤1M KRW	136	66.58 ± 17.12 (63.85–69.32)	96.94 ± 11.05 (94.88–98.99)	66.05 ± 14.88 (63.50–68.61)	65.93 ± 14.88 (63.02–68.84)
>1M, ≤3M	78	63.46 ± 17.46) (59.85–67.08)	96.69 ± 10.93 (93.98–99.40)	65.74 ± 15.92 (62.36–69.12)	62.96 ± 17.49 (59.12–66.81)
>3M	51	55.77 ± 10.77 *^,‡^ (51.30–60.24)	93.30 ± 16.16 (89.94–96.65)	62.31 ± 14.58 (58.13–66.48)	55.34 ± 14.82 *^,‡^ (50.58–60.09)
**Comorbidity (Prob)**		0.0616	0.6804	0.2997	0.1447
0	221	63.12 ± 16.07 (60.87–65.37)	95.63 ± 12.65 (93.97–97.28)	65.13 ± 14.65 (63.11–67.15)	61.31 ± 18.06 (58.91–64.72)
1	55	68.89 ± 16.37 (64.37–73.40)	97.27 ± 10.88 (93.95–100.60)	67.47 ± 15.13 (63.43–71.52)	65.81 ± 18.57 (60.98–70.63)
≥2	37	66.52 ± 22.66 (61.01–72.02)	96.17 ± 13.97 (92.12–100.22)	62.46 ± 18.63 (57.53–67.40)	65.69 ± 18.31 (59.81–71.57)
**Treatment duration** **(Prob)**		0.0017	0.3472	0.0289	0.0015
<1 year	17	58.17 ± 14.31 (50.17–66.17)	90.20 ± 17.73 (84.23–96.16)	63.73 ± 15.72 (56.52–70.93)	53.27 ± 22.44 (44.75–61.78)
≥1, <3 years	35	61.90 ± 15.19 (56.33–67.48)	97.86 ± 9.33 (93.70–102.01)	68.25 ± 11.39 (63.23–73.27)	62.70 ± 15.94 (56.77–68.63)
≥3, <5 years	54	59.16 ± 13.70 (54.67–63.64)	96.30 ± 11.17 (92.95–99.64)	63.68 ± 15.99 (59.64–67.72)	57.82 ± 16.72 (53.04–62.60)
≥5, <10 years	60	62.96 ± 18.60 (58.71–67.22)	96.11 ± 11.73 (92.94–99.28)	60.37 ± 15.76 (56.53–64.20)	59.54 ± 18.77 (55.01–64.07)
≥10 years	147	68.52 ± 17.54 ^‡^ (65.80–71.24)	96.03 ± 13.17 (94.00–98.06)	67.23 ± 15.16 ^※^ (64.78–69.68)	66.70 ± 17.68 *^,‡^ (64.81–69.60)
**Monthly Drug OOP** **(Prob)**		0.0177	0.5310	0.1537	0.1983
<5K KRW	73	70.24 ± 21.10 (66.34–74.14)	97.15 ± 10.86 (94.25–100.04)	67.28 ± 15.56 (63.78–70.77)	66.25 ± 20.40 (62.05–70.44)
≥5K, <10K	47	63.83 ± 15.36 (58.97–68.69)	95.74 ± 11.50 (92.14–99.35)	66.67 ± 12.53 (62.31–71.03)	65.01 ± 16.60 (59.78–70.24)
≥10K, <20K	79	61.53 ± 13.80 * (57.78–65.28)	94.62 ± 15.15 (91.84–97.40)	66.39 ± 13.34 (63.02–69.75)	60.72 ± 17.00 (56.69–64.76)
≥20K, <30K	48	64.93 ± 15.16 (60.12–69.74)	97.92 ± 7.78 (94.35–101.48)	61.69 ± 16.81 (57.38–66.00)	59.90 ± 16.76 (54.72–65.07)
≥30K	64	62.15 ± 17.42 * (57.99–66.32)	94.92 ± 14.28 (91.83–98.00)	62.59 ± 17.26 (58.85–66.32)	61.15 ± 19.14 (56.67–65.63)

Note: * significant difference compared with the first subgroup (*p* < 0.05); ^†^ significant difference compared with the second subgroup (*p* < 0.05); ^‡^ significant difference compared with the third subgroup (*p* < 0.05); and ^※^ significant difference compared with the fourth subgroup (*p* < 0.05). CI: confidence interval; OOP: Out of Pocket; KRW: Korean Won.

**Table 5 healthcare-13-02918-t005:** Correlation between PMRRs and the TSQM by dimensions (*n* = 313).

PMRR Domain	TSQM Domain			
	Effectiveness	Side Effects	Convenience	Global Satisfaction
Effectiveness	0.451 **/0.431 **	0.145 */0.142 *	0.240 **/0.234 **	0.384 **/0.369 **
Side Effects	0.400 **/0.391 **	0.364 **/0.362 **	0.133 */0.134 *	0.354 **/0.347 **
Ease of Use	0.318 **/0.312 **	0.078/0.081	0.286 **/0.283 **	0.321 **/0.320 **
Affordability	0.233 **/0.233 **	0.077/0.069	0.145 */0.130 *	0.222 **/0.207 *
Food interaction	0.307 **/0.298 **	0.137 */0.144 *	0.130 */0.124 *	0.324 **/0.326 **
Willingness to Recommend	0.271 **/0.274 **	0.193 */0.186 *	0.120 */0.127 *	0.256 **/0.244 **

Note: Analysis performed with Pearson correlation, and Pearson partial correlation and coefficients are shown with ‘Unadjusted/Adjusted’. Participants’ characteristics with sex, age, education, comorbidities, and polypharmacy have been used for adjustment; * *p* < 0.05; ** *p* < 0.0001.

## Data Availability

The data presented in this study are available upon request from the corresponding author. The data are not publicly available due to privacy restrictions.

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
