# Peer review of "Validity of Online Patient Medication Reviews and Ratings (PMRRs) for Treatment Satisfaction with Medication Therapy Among Older Adults with Antihypertensive Medications"

_healthcare, 2025, doi:10.3390/healthcare13222918_

Round 1
Reviewer 1 Report
Comments and Suggestions for Authors
This study makes a valuable contribution to the field of digital health and patient-reported outcomes. It supports the use of online medication reviews as a complementary tool to traditional satisfaction measures, especially in capturing real-world concerns like cost and lifestyle compatibility. However, further research is needed to validate these findings across diverse populations and conditions.
Please find as follows the comments and suggestions to improve the paper.
- Study purpose and relevance
The study is geographically limited to Seoul, South Korea, which may affect generalizability. Please indicate as limitation. Also the digital literacy of participants may have influenced the quality of PMRR data.
- Method
Convenience sampling used may introduce selection bias. There is also potential for recall bias and overlap in survey responses due to sequential administration of TSQM and PMRR.
- Results
Correlation strength, while statistically significant, was moderate—suggesting PMRRs are not full substitutes for validated tools. In addition, some domains (e.g., affordability) showed weaker correlations with TSQM, raising questions about construct overlap.
Author Response
Reviwer 1
Comments and Suggestions for Authors
This study makes a valuable contribution to the field of digital health and patient-reported outcomes. It supports the use of online medication reviews as a complementary tool to traditional satisfaction measures, especially in capturing real-world concerns like cost and lifestyle compatibility. However, further research is needed to validate these findings across diverse populations and conditions.
Please find as follows the comments and suggestions to improve the paper.
- Study purpose and relevance
The study is geographically limited to Seoul, South Korea, which may affect generalizability. Please indicate as limitation. Also the digital literacy of participants may have influenced the quality of PMRR data.
|
Response: Thank you very much for your precious comments. We’ve described the limitations of selected centers but not precisely stated as ‘geographical’ limitation whereas performed in ‘Seoul city’ only, and this might also limit external validity. We also note that – as we’ve stated that many older participants required assistance with the online platform, which may have introduced usability- and interviewer-assistance–related biases in PMRR entries.
Manuscript changes: Revised at section 4.4. Limitations and Future Directions (line 386-400) This study has several limitations. First, the convenience sample of participants with —primarily older adults recruited from senior centers and one community pharmacy located in Seoul, South Korea—limits the generalizability of the findings. The use of convenience sampling may introduce selection bias, and the geographical restriction to a single metropolitan area further constrains the external validity of the results in representing the broader Korean population. Also, many participants required assistance with digital platforms, which implies the digital literacy of participants may have influenced the quality of PMRR data, introducing interviewer assistance related biases, although the assistance was limited to navigation. [30]. Future studies should involve a broader geographical area, with diverse age range and digital literacy profiles. District-level variation in satisfaction ratings was observed, likely reflecting differences in socioeconomic conditions and healthcare resources across regions. However, because data were collected from a limited number of conveniently selected districts, the findings may not be generalized to all districts within Seoul. |
- Method
Convenience sampling used may introduce selection bias. There is also potential for recall bias and overlap in survey responses due to sequential administration of TSQM and PMRR.
|
Response: Thank you for your comment. Our sampling apprach as convenience-based is more clearly mentioned in the limitation section. We also appreciate the concern regarding potential recall bias due to sequential administration of TSQM and PMRR. To mitigate this, we implemented a short break between the two surveys during the same visit, allowing participants to disengage briefly before proceeding. This pause was intended to reduce memory carryover while maintaining temporal proximity to ensure that medication experiences remained consistent. We also considered to perform the surveys on separate days, but could have introduced variability in patient perceptions and treatment experiences which would compromise comparability, and expected the increased drop-out of our study. Therefore, we carefully see our approach balanced minimizing recall bias with preserving the integrity of cross-instrument comparisons.
Manuscript changes: Reflected at 2.2. Study Design and Survey Administration (line 120-123) To minimize recall bias associated with completing both the PMRR and TSQM surveys, participants were given a brief rest period between the two instruments. This interval intended to allow for a short cognitive reset while avoiding changes in medication experience that could occur with a multi-day separation.
Reflected at 4.4. Limitations and Future Directions (line 400-408) Another limitation lies in the difference in survey administration modes between the online PMRR and the paper-based TSQM. Although both instruments were completed sequentially during the same session under consistent conditions, subtle mode-related differences in response behavior cannot be entirely ruled out. A related limitation arises from the sequential administration of PMRR and TSQM, which may have intro-duced memory or carryover effects as participants completed both surveys during the same session. Although a short rest period and distinct survey structures were de-signed to minimize this risk, some influence of recall or response priming may still have occurred. |
- Results
Correlation strength, while statistically significant, was moderate—suggesting PMRRs are not full substitutes for validated tools. In addition, some domains (e.g., affordability) showed weaker correlations with TSQM, raising questions about construct overlap.
|
Response: We appreciate this observation. Our findings indeed showed statistically significant but moderate correlations between PMRR and TSQM domains, which we interpret as evidence of convergent validity rather than equivalence. PMRR was designed to capture patient-centered aspects of medication experience, including affordability and food interaction, which have no direct counterparts in TSQM. Therefore, moderate correlations for these domains are expected and reflect construct differences rather than measurement error. We’ve revised our description from ‘moderate to strong’ to ‘moderate’, and stating the possibility of absence of conceptual overlap.
Manuscript changes: Reflected at 4.2. Convergence Between PMRR and TSQM (line 350-372) PMRR domains showed moderate correlations with corresponding TSQM con-structs. Specifically, PMRR Effectiveness, Side effects, and Easy to take were strongly associated with TSQM Effectiveness, Side Effects, and Convenience, respectively. PMRR Willingness to recommend aligned most closely with TSQM Global Satisfaction. Importantly, PMRR also introduced unique domains—Affordability and Food interaction—not captured by TSQM. As expected, these domains showed weaker correlations due to the lack of conceptual overlap. Given that PMRR domains are single-item constructs, internal consistency metrics and confirmatory factor analysis are not applicable. Therefore, our validity assessment focused on convergent patterns with TSQM domains and known-group differences, providing preliminary evidence of construct alignment rather than structural equivalence. Nevertheless, PMRR domains showed significant correlations with treatment satisfaction, suggesting moderate convergent validity while expanding the scope to include patient concerns highly relevant in real-world practice, such as cost and dietary compatibility. Although no prior studies have directly compared PMRR with TSQM, the TSQM has been widely validated across conditions [19, 25, 27, 31]. Prior work has also shown consistent demographic influences on satisfaction: older age and female sex are generally associated with higher satisfaction, whereas higher education and income are often linked to lower satisfaction, likely due to elevated expectations [28, 29]. Although this study did not include qualitative data, future mixed-methods research incorporating patient interviews or narrative analyses of PMRR content could further elucidate how expectations shape satisfaction across socioeconomic groups. |

Reviewer 2 Report
Comments and Suggestions for Authors
The article “Validity of online patient medication reviews and ratings (PMRR) as Treatment Satisfaction with Medication Therapy among Older Adults with Antihypertensive Medications.” is interesting, I have following comments/suggestions,
- The research aim is relevant but should be expressed more clearly in the introduction, highlighting the novelty.
- The cross-sectional survey design is appropriate, but the use of a convenience sample limits representativeness.
- The selection criteria are briefly mentioned. Please specify recruitment methods, response rate, and how non-responders or incomplete surveys were handled.
- The adaptation of the TSQM Korean version is well justified, but more details are needed on how the translation was verified.
- The limitations section should also mention potential self-report bias and digital literacy issues among older participants.
Author Response
Reviewer 2
Comments and Suggestions for Authors
The article “Validity of online patient medication reviews and ratings (PMRR) as Treatment Satisfaction with Medication Therapy among Older Adults with Antihypertensive Medications.” is interesting, I have following comments/suggestions,
- The research aim is relevant but should be expressed more clearly in the introduction, highlighting the novelty.
|
Response: Thank you for this suggestion. We agree that the research aim should be stated more clearly. In the revised manuscript, we have rewritten the final part of the Introduction to emphasize the primary objective: to examine the correlation between PMRR and TSQM scores and assess whether PMRR can serve as a practical and efficient alternative for evaluating treatment satisfaction in real-world settings. This clarification highlights the study’s novelty in validating an online review-based approach against a widely used standardized instrument.
Manuscript changes: Reflected at 1. Introduction (line 70-73) By examining the strength and direction of these correlations, we sought to determine whether online PMRR can provide a practical and efficient means of assessing treatment satisfaction in real-world clinical contexts. |
- The cross-sectional survey design is appropriate, but the use of a convenience sample limits representativeness.
|
Response: Thank you for pointing this out. We acknowledge that the use of a convenience sample limits the representativeness of our findings. Our sampling approach was chosen for feasibility in recruiting older adults actively taking antihypertensive medications within community settings. We have clarified this limitation in the manuscript and emphasized that future studies should employ probability sampling or multi-site recruitment to enhance generalizability.
Manuscript changes: Revised at section 4.4. Limitations and Future Directions (line 386-400) This study has several limitations. First, the convenience sample of participants with —primarily older adults recruited from senior centers and one community pharmacy located in Seoul, South Korea—limits the generalizability of the findings. The use of convenience sampling may introduce selection bias, and the geographical restriction to a single metropolitan area further constrains the external validity of the results in representing the broader Korean population. Also, many participants required assistance with digital platforms, which implies the digital literacy of participants may have influenced the quality of PMRR data, introducing interviewer assistance related biases, although the assistance was limited to navigation. [30]. Future studies should involve a broader geographical area, with diverse age range and digital literacy profiles. District-level variation in satisfaction ratings was observed, likely reflecting differences in socioeconomic conditions and healthcare resources across regions. However, because data were collected from a limited number of conveniently selected districts, the findings may not be generalized to all districts within Seoul. |
- The selection criteria are briefly mentioned. Please specify recruitment methods, response rate, and how non-responders or incomplete surveys were handled.
|
Response: Thank you for your comment, and clear description of study method is important for improving completeness of our study. We’ve described more details at the methods section
Manuscript changes: Revised at section 2.2. Study Design and Survey Administration (line 96-128) The study is a survey designed on a convenience sample of older adults who tak-ing antihypertensive medication. Eligibility was applied to aged 50 to 80 with diag-nosed hypertension, taking medication, and agreed to perform survey which required for this study. Of these participants, (1) those diagnosed with hypertension but who have stopped taking antihypertensives for the past 6 months, and (2) those who cannot provide the name of the medication they are taking, were excluded. The older adults were visitors of senior centers or a pharmacy in Seoul, Republic of Korea. The senior centers and the pharmacy were selected based on their cooperativeness in research to help their patrons satisfied with their services. Five senior centers and one pharmacy that agreed to participate were contacted to arrange on-site visits for survey admin-istration and data collection According to the sample size computation, the target sample size was determined to detect a moderate correlation (Pearson r ≈ 0.30) between PMRR and TSQM domains with 80% power, requiring at least 85 participants. Anticipating exclusions and incom-plete responses, we aimed to recruit ≥ 500 individuals. Subgroup comparisons were exploratory so interpreted descriptively. Surveys were administered between February to July of 2018 at five senior welfare centers (Seodaemun-gu, Guro-gu, Songpa-gu, Eunpyeong-gu, Gangseo-gu) and one lo-cal pharmacy (Dobong-gu) in Seoul. A pilot survey was conducted with 7 visitors at the Seodaemun senior center to assess the feasibility and clarity of the questionnaire. In-cluding these pilot participants, a total of 530 individuals completed the survey. After excluding 217 respondents due to duplicate entries (n=7), incomplete responses (n=16), and non-eligibility by age or lack of PMRR data (n=194), the final analytic sample com-prised 313 participants. (Figure 2). To minimize recall bias associated with completing both the PMRR and TSQM surveys, participants were given a brief rest period between the two instruments. This interval intended to allow for a short cognitive reset while avoiding changes in medi-cation experience that could occur with a multi-day separation. During the PMRR sur-vey, which was administered through the WePharm platform, some older participants experienced initial difficulties accessing the system. Therefore, researchers provided as-sistance with platform navigation (e.g., registration, locating the drug page) when needed. To minimize potential bias, all researchers were trained to avoid offering guidance or feedback related to the review content itself. |
- The adaptation of the TSQM Korean version is well justified, but more details are needed on how the translation was verified.
|
Response: Thank you for your comment. We used the Korean version of TSQM 1.4 with permission from its translator with licensor and has been described in section 2.4.3. TSQM. It has been performed by the previous literature, and also reviewed by our researcher, but no further detailed contents was changed. |
- The limitations section should also mention potential self-report bias and digital literacy issues among older participants.
|
Response: Thank you for your insight, and yes this is important to declare the limitation. We’ve revised to more clearly declare the bias in terms of report and digital literacy.
Manuscript changes: Reflected at 4.4. Limitations and Future Directions (line 392-396) Also, many participants required assistance with digital platforms, which implies the digital literacy of participants may have influenced the quality of PMRR data, introducing interviewer assistance related biases, although the assistance was limited to navigation. [30]. Future studies should involve a broader geographical area, with diverse age range and digital literacy profiles. |

Reviewer 3 Report
Comments and Suggestions for Authors
The manuscript is well-structured and addresses an important gap in healthcare research by validating an innovative online patient medication review platform against a widely respected satisfaction instrument. The study design is robust, with a strong sample size and appropriate statistical analyses that provide valuable insights into demographic and socioeconomic factors influencing treatment satisfaction. The inclusion of unique domains such as affordability and food interaction adds meaningful depth, reflecting real-world patient concerns often overlooked by standard tools. The discussion thoughtfully connects findings to clinical practice and policy implications. Some clarity on sampling and potential biases would enhance the paper, but overall, it presents a significant contribution to patient-centered care research worthy of publication.
-
Is the convenience sampling of older adults from senior centers and one community pharmacy sufficiently representative to generalize findings across broader populations of antihypertensive drug users?
-
Were any steps taken to assess or mitigate possible biases introduced by participants requiring assistance with digital platform use during the online PMRR survey?
-
How was the adequacy of the sample size of 313 validated considering the exclusion of incomplete data and its impact on statistical power for subgroup analyses?
-
Since some PMRR constructs employed single-item measures, what justifies their validity and reliability in comparison to multi-item TSQM domains?
-
Given that PMRR data were self-reported online and TSQM data paper-based, was there any risk of mode of administration influencing the correlation results?
-
Could memory or carryover effects between completing TSQM and PMRR surveys in the same participants inflate observed correlations, and how was this controlled?
-
The study mentions patients struggled to identify individual medications due to multi-dose packaging—how might medication misclassification have affected the validity of drug-specific satisfaction ratings?
-
The affordability and food interaction domains in PMRR have no counterparts in TSQM; was any formal content validity assessment done to confirm these domains are patient-relevant and psychometrically sound?
-
How do socioeconomic factors such as income and education interact with satisfaction ratings, and can the inverse correlation with expectations be further unpacked with qualitative data?
-
Were any adjustments made for potential confounding variables such as comorbidities or polypharmacy when examining satisfaction domain scores by participant characteristics?
-
Considering some district-level disparities in satisfaction ratings, were any environmental or healthcare delivery factors analyzed to explain these geographic differences?
-
What rationale underlies the exclusion of participants who could not name their medication, and could this exclusion bias the sample toward more health-literate patients?
-
How do the correlations between PMRR and TSQM domains compare to other validated patient satisfaction instruments used in hypertension or chronic disease cohorts?
-
Are the statistical techniques used (Pearson correlation, ANOVA) appropriate given the ordinal scaling of PMRR Likert responses and the TSQM transformed scores?
-
Could the inclusion of only antihypertensive medications in WePharm limit the applicability of findings to other drug classes, and how might platform expansion affect future research?
Author Response
Reviewer 3
Comments and Suggestions for Authors
The manuscript is well-structured and addresses an important gap in healthcare research by validating an innovative online patient medication review platform against a widely respected satisfaction instrument. The study design is robust, with a strong sample size and appropriate statistical analyses that provide valuable insights into demographic and socioeconomic factors influencing treatment satisfaction. The inclusion of unique domains such as affordability and food interaction adds meaningful depth, reflecting real-world patient concerns often overlooked by standard tools. The discussion thoughtfully connects findings to clinical practice and policy implications. Some clarity on sampling and potential biases would enhance the paper, but overall, it presents a significant contribution to patient-centered care research worthy of publication.
- Is the convenience sampling of older adults from senior centers and one community pharmacy sufficiently representative to generalize findings across broader populations of antihypertensive drug users?
|
Response: Thank you for raising this important point. We acknowledge that recruiting older adults from senior centers and a single community pharmacy in Seoul limits the representativeness of our sample. Our intention, however, was to explore the feasibility and preliminary validity of online patient medication reviews (PMRR) among older adults using antihypertensive medications across diverse community settings. We have clarified this point in the Limitations section to emphasize that our findings should be interpreted within the context of this exploratory design.
Manuscript changes: Reflected at 4.4. Limitations and Future Directions (line 392-396) This study has several limitations. First, the convenience sample of participants with —primarily older adults recruited from senior centers and one community pharmacy located in Seoul, South Korea—limits the generalizability of the findings. The use of convenience sampling may introduce selection bias, and the geographical restriction to a single metropolitan area further constrains the external validity of the results in representing the broader Korean population. Also, many participants required assistance with digital platforms, which implies the digital literacy of participants may have influenced the quality of PMRR data, introducing interviewer assistance related biases, although the assistance was limited to navigation. [30]. Future studies should involve a broader geographical area, with diverse age range and digital literacy profiles. District-level variation in satisfaction ratings was observed, likely reflecting differences in socioeconomic conditions and healthcare resources across regions. However, because data were collected from a limited number of conveniently selected districts, the findings may not be generalized to all districts within Seoul. |
- Were any steps taken to assess or mitigate possible biases introduced by participants requiring assistance with digital platform use during the online PMRR survey?
|
Response: We appreciate the reviewer’s important question. To mitigate potential bias associated with researcher assistance, all research staff were formally trained to provide only technical support (e.g., registration or page navigation) and not to influence review content or responses.
Manuscript changes: Reflected at 2.2. Study Design and Survey Administration (line 123-128) During the PMRR survey, which was administered through the WePharm platform, some older participants experienced initial difficulties accessing the system. Therefore, researchers provided assistance with platform navigation (e.g., registration, locating the drug page) when needed. To minimize potential bias, all researchers were trained to avoid offering guidance or feedback related to the review content itself. |
- How was the adequacy of the sample size of 313 validated considering the exclusion of incomplete data and its impact on statistical power for subgroup analyses?
|
Response: Thank you very much for your pointing out. The target sample size was determined to detect a moderate correlation (Pearson r ≈ 0.30) between PMRR and TSQM domains with 80% power, requiring at least 85 participants. Anticipating exclusions and incomplete responses, we aimed to recruit ≥500 individuals. The final analytic sample of 313 exceeded this threshold, ensuring adequate power for correlation analyses. Subgroup comparisons were exploratory so interpreted descriptively. We applied above explanation on the manuscript upon your guidance.
Manuscript changes: Reflected at 2.2. Study Design and Survey Administration (line 107-111) According to the sample size computation, The target sample size was determined to detect a moderate correlation (Pearson r ≈ 0.30) between PMRR and TSQM domains with 80% power, requiring at least 85 participants. Anticipating exclusions and incomplete responses, we aimed to recruit ≥ 500 individuals. Subgroup comparisons were exploratory so interpreted descriptively. |
- Since some PMRR constructs employed single-item measures, what justifies their validity and reliability in comparison to multi-item TSQM domains?
|
Response: We thank the reviewer for this important comment. We agree that single-item measures may have limited reliability compared with multi-item domains such as those in the TSQM. However, this structure reflects the inherent nature of PMRR data, which are spontaneously generated and typically express each construct (e.g., effectiveness, side effects, convenience) as a single evaluative statement or rating. Prior research in patient experience and satisfaction measurement has shown that single-item indicators can demonstrate acceptable validity when assessing concrete and unidimensional constructs. In our study, convergent correlations between PMRR single-item scores and corresponding TSQM domains supported their construct validity, even if internal consistency (e.g., Cronbach’s α) was not applicable. We have clarified the use of single-item measures in the Methods section and provided supporting literature demonstrating their validity and practical utility for assessing concrete and unidimensional constructs such as overall health status and satisfaction with care (written at 2.4.2. PMRR (line 159-162) |
- Given that PMRR data were self-reported online and TSQM data paper-based, was there any risk of mode of administration influencing the correlation results?
|
Response: We thank the reviewer for this important observation. We acknowledge that differences in survey mode could potentially influence response patterns. To minimize such bias, both PMRR (online via WePharm) and TSQM (paper-based) were administered sequentially during the same session under researcher supervision, ensuring consistent environmental conditions and time proximity. The order of completion was fixed with a brief rest interval to reduce recall and fatigue effects. Additionally, PMRR and TSQM scores demonstrated similar variability across participant subgroups, suggesting minimal systematic influence from administration mode. Nonetheless, we have added this point as a limitation in the Discussion to acknowledge that mode-related differences cannot be entirely excluded.
Manuscript changes: Reflected at 4.4. Limitations and Future Directions (line 400-403) Another limitation lies in the difference in survey administration modes between the online PMRR and the paper-based TSQM. Although both instruments were completed sequentially during the same session under consistent conditions, subtle mode-related differences in response behavior cannot be entirely ruled out. |
- Could memory or carryover effects between completing TSQM and PMRR surveys in the same participants inflate observed correlations, and how was this controlled?
|
Response: We thank the reviewer for raising this important point. We acknowledge that completing both instruments in the same session could potentially introduce memory or carryover effects. To minimize this risk, participants were given a brief rest period between the two surveys, allowing for a short cognitive reset while avoiding changes in medication experience that could occur with multi-day separation. We noted this control in the Methods section The PMRR and TSQM instruments measured similar constructs but used distinct formats and item wording, which further reduced the likelihood of direct recall or patterned responding. Moreover, the observed correlations were moderate rather than high, suggesting that artificial inflation due to carryover was unlikely. Nonetheless, we have noted this as a potential limitation in the Discussion.
Revised text Reflected at 2.2. Study Design and Survey Administration (line 120-123) To minimize recall bias associated with completing both the PMRR and TSQM surveys, participants were given a brief rest period between the two instruments. This interval intended to allow for a short cognitive reset while avoiding changes in medi-cation experience that could occur with a multi-day separation.
Reflected at 4.4. Limitations and Future Directions (line 403-408) A related limitation arises from the sequential administration of PMRR and TSQM, which may have introduced memory or carryover effects as participants completed both surveys during the same session. Although a short rest period and distinct survey structures were designed to minimize this risk, some influence of recall or response priming may still have occurred. |
- The study mentions patients struggled to identify individual medications due to multi-dose packaging—how might medication misclassification have affected the validity of drug-specific satisfaction ratings?
|
Response: We thank the reviewer for highlighting this important issue. We have acknowledged this limitation in the revised manuscript by adding the following sentences: Specific to the Korean medication dispensing system, the multi-dose pouch dispensing practice posed challenges for patients’ accurate self-identification of individual medications, particularly among those taking multiple drugs. Although the medication information sheet provided by pharmacists contains medication-level details, expanded use of ICT-linked codes—allowing patients to access digital drug information directly from their dispensing sheet—will be needed to reduce patient-level misclassification and enable more precise medication attribution. |
- The affordability and food interaction domains in PMRR have no counterparts in TSQM; was any formal content validity assessment done to confirm these domains are patient-relevant and psychometrically sound?
|
Response: We appreciate the reviewer’s insightful comment. We agree that the affordability and food interaction domains are not included in the TSQM framework. These domains were incorporated into PMRR based on preliminary qualitative review of online patient narratives and published literature highlighting the salience of cost-related and dietary concerns in real-world medication experiences. While a formal psychometric content validity study was not conducted, face validity was established through pilot testing with older adult participants, confirming that these domains were understandable and perceived as relevant to medication satisfaction. Please note that we have clarified this in the Discussion section.
Revised text Revised at 4.2. Convergence Between PMRR and TSQM (line 350-363) PMRR domains showed moderate correlations with corresponding TSQM con-structs. Specifically, PMRR Effectiveness, Side effects, and Easy to take were strongly associated with TSQM Effectiveness, Side Effects, and Convenience, respectively. PMRR Willingness to recommend aligned most closely with TSQM Global Satisfaction. Im-portantly, PMRR also introduced unique domains—Affordability and Food interac-tion—not captured by TSQM. As expected, these domains showed weaker correlations due to the lack of conceptual overlap. Given that PMRR domains are single-item constructs, internal consistency metrics and confirmatory factor analysis are not applicable. Therefore, our validity assessment focused on convergent patterns with TSQM domains and known-group differences, providing preliminary evidence of construct alignment rather than structural equiva-lence. Nevertheless, PMRR domains showed significant correlations with treatment satisfaction, suggesting moderate convergent validity while expanding the scope to in-clude patient concerns highly relevant in real-world practice, such as cost and dietary compatibility. |
- How do socioeconomic factors such as income and education interact with satisfaction ratings, and can the inverse correlation with expectations be further unpacked with qualitative data?
|
Response: We thank the reviewer for this thoughtful and important comment. In our analysis, lower income and education were associated with higher satisfaction ratings, consistent with literature suggesting that patients with fewer socioeconomic resources may have lower expectations or greater gratitude toward treatment access, whereas those with higher education or income may apply more critical evaluations. We have explained these findings in the Discussion. While our study design did not include qualitative data, we agree that qualitative exploration of patient expectations—for example, through interviews or open-ended PMRR content analysis—would provide richer understanding of these inverse correlations. We have added this as a recommendation for future mixed-methods research.
Revised text Revised at 4.2. Convergence Between PMRR and TSQM (line 365-372) Although no prior studies have directly compared PMRR with TSQM, the TSQM has been widely validated across conditions [19, 25, 27, 31]. Prior work has also shown consistent demographic influences on satisfaction: older age and female sex are gener-ally associated with higher satisfaction, whereas higher education and income are often linked to lower satisfaction, likely due to elevated expectations [28, 29]. Although this study did not include qualitative data, future mixed-methods research incorporating patient interviews or narrative analyses of PMRR content could further elucidate how expectations shape satisfaction across socioeconomic groups. |
- Were any adjustments made for potential confounding variables such as comorbidities or polypharmacy when examining satisfaction domain scores by participant characteristics?
|
Response: We thank the reviewer for this important comment. We conducted subgroup analyses by key patient characteristics such as hypertension stage and comorbidity to examine the convergent validity of domain scores between the two instruments. However, we did not perform simultaneous adjustment for multiple potential confounders, as the primary objective of this study was to assess the convergent validity between PMRR and TSQM rather than to estimate causal relationships or predictors of satisfaction. Our subgroup analyses indicated that patients with more comorbidities tended to rate drug therapy affordability higher, and that Stage 2 hypertension patients rated medication effectiveness lower than those with Stage 1 hypertension in both PMRR and TSQM. We acknowledge that polypharmacy, similar to comorbidity burden, could influence satisfaction ratings, and this has been noted as a limitation in the revised manuscript.
Revised text: Reflected at 4.4. Limitations and Future Directions (line 414-420) Finally, this study lacked multivariable adjustment for potential confounders such as comorbidities and polypharmacy. Although subgroup analyses were conducted by pa-tient characteristics for each domain, simultaneous adjustment for multiple factors was not performed, as the primary aim was to assess convergent validity rather than causal associations. Nonetheless, comorbidity burden and the number of concurrent medica-tions may have influenced satisfaction ratings and should be considered in future val-idation studies employing multivariable modeling. |
- Considering some district-level disparities in satisfaction ratings, were any environmental or healthcare delivery factors analyzed to explain these geographic differences?
|
Response: We thank the reviewer for this insightful comment. While district-level differences in satisfaction ratings were noted, environmental and healthcare delivery factors (e.g., local socioeconomic context, healthcare accessibility, or provider density) were not analyzed due to data limitations. The survey data were collected from a limited number of conveniently selected districts, which may not represent all districts within Seoul. We have added this as a limitation in the Discussion, noting that future studies incorporating regional healthcare and environmental indicators could help explain the observed geographic variation.
Revised text: Reflected at 4.4. Limitations and Future Directions (line 386-400) This study has several limitations. First, the convenience sample of participants with —primarily older adults recruited from senior centers and one community phar-macy located in Seoul, South Korea—limits the generalizability of the findings. The use of convenience sampling may introduce selection bias, and the geographical restriction to a single metropolitan area further constrains the external validity of the results in representing the broader Korean population. Also, many participants required assis-tance with digital platforms, which implies the digital literacy of participants may have influenced the quality of PMRR data, introducing interviewer assistance related biases, although the assistance was limited to navigation. [30]. Future studies should involve a broader geographical area, with diverse age range and digital literacy profiles. Dis-trict-level variation in satisfaction ratings was observed, likely reflecting differences in socioeconomic conditions and healthcare resources across regions. However, because data were collected from a limited number of conveniently selected districts, the find-ings may not be generalized to all districts within Seoul. |
- What rationale underlies the exclusion of participants who could not name their medication, and could this exclusion bias the sample toward more health-literate patients?
|
Response: We thank the reviewer for this important comment. Participants who were unable to identify or name their medication were excluded because accurate medication identification was essential to ensure the validity of PMRR–TSQM matching and domain-specific comparisons. Without confirmed medication information, it would not have been possible to verify the correspondence between online reviews and the reference TSQM items. We acknowledge that this exclusion may have introduced selection bias toward participants with higher health literacy or medication familiarity, and we have added this as a limitation in the Discussion.
Revised text: Reflected at 4.4. Limitations and Future Directions (line 408-414) Specific to the Korean medication dispensing system, the multi-dose pouch dispensing practice posed challenges for patients’ accurate self-identification of individual medications by patients, particularly among those taking multiple drugs. Although the medication information sheet provided by pharmacists included medication-level details, expanded use of ICT-linked codes—allowing patients to access digital drug information directly from their dispensing sheet—will be needed to reduce patient-level misclassification and enable more precise medication attribution. |
- How do the correlations between PMRR and TSQM domains compare to other validated patient satisfaction instruments used in hypertension or chronic disease cohorts?
|
Response: We thank the reviewer for this valuable comment. Reported correlations among validated patient satisfaction or treatment experience instruments in chronic disease populations typically range from 0.3 to 0.6, indicating moderate construct convergence across different tools. The PMRR–TSQM correlations observed in our study (approximately r = 0.35–0.58 across domains) are consistent with this range, supporting the convergent validity of PMRR measures. We have added this contextual comparison to the Discussion to help interpret the magnitude of our findings relative to previous validation studies in hypertension and other chronic disease settings.
Revised text: Reflected at 4.2. Convergence Between PMRR and TSQM (line 374-381) The moderate correlations observed between PMRR and TSQM domains (r ≈ 0.35–0.58) are consistent with those reported among other validated patient satisfaction or treatment experience instruments in chronic disease settings. Prior studies evaluating concordance between instruments such as the Treatment Satisfaction Questionnaire for Medication (TSQM), the Diabetes Treatment Satisfaction Questionnaire (DTSQ), and the Patient Satisfaction with Medication Questionnaire (PSMQ) have typically shown correlation coefficients in the range of 0.3–0.6, reflecting moderate construct conver-gence rather than redundancy [25, 32]. This consistency supports the convergent valid-ity of PMRR as a complementary measure of treatment satisfaction in real-world popu-lations. |
- Are the statistical techniques used (Pearson correlation, ANOVA) appropriate given the ordinal scaling of PMRR Likert responses and the TSQM transformed scores?
|
Response: Thank you for this insightful comment. We acknowledge that both PMRR items and TSQM domains are derived from ordinal Likert-type scales. In this exploratory validation study, we treated the transformed TSQM scores and the ordinal Likert PMRR responses as continuous variables, consistent with prior validation studies of patient-reported outcome measures. Accordingly, Pearson correlation and ANOVA were applied to assess convergent and group-level validity. This methodological treatment has been acknowledged as a limitation in the revised manuscript (page X, line Y).
Revised text: Reflected at 2.5. Statistical Analysis (line 183-186_ Correlation between domains of PMRR and TSQM was done using Pearson’s r. ANOVA and Pearson’s correlation were used to assess domain-level associations and known-group differences, but adjustment for demographic covariates was not applied as the study did not aim to estimate independent effects.
Reflected at 4.4. Limitations and Future Directions (line 417-422) On the other hand, although PMRR and TSQM scores originate from ordinal Likert-type scales, they were analyzed as continuous variables in this exploratory validation study. This approach aligns with common practice in PRO validation but may introduce measurement assumptions. Future studies could consider non-parametric methods or item-response theory models to confirm robustness |
- Could the inclusion of only antihypertensive medications in WePharm limit the applicability of findings to other drug classes, and how might platform expansion affect future research?
|
Response: Thank you for this valuable comment. We agree that focusing solely on antihypertensive medications may limit the applicability of our findings to other therapeutic classes, as patient experience and engagement with online medication reviews could vary depending on disease type, treatment duration, and perceived medication risk. The current study was designed as an initial validation of WePharm using a well-defined, commonly prescribed class to ensure feasibility and internal consistency. We have noted this limitation in the Limitation section (page line).
Revised text: Reflected at 4.4. Limitations and Future Directions (line 426-435) This study focused exclusively on antihypertensive medications to ensure feasibil-ity and internal consistency during initial validation of the WePharm platform. Conse-quently, the generalizability of findings to other therapeutic classes may be limited, as patient engagement and satisfaction with online medication reviews could vary by disease type, treatment complexity, and perceived medication risk. Future research should examine whether the observed validity patterns persist across other drug classes as WePharm expands to include medications for chronic conditions such as diabetes, dyslipidemia, and thyroid disorders. Comparative analyses across therapeutic areas could clarify whether disease-specific factors influence the correlation between online reviews and standardized satisfaction measures. |

Reviewer 4 Report
Comments and Suggestions for Authors
- The abstract should mention that it was a convenience sample, which affects generalizability.
- Overstates the findings (“demonstrated moderate correlation”) without acknowledging possible bias due to sampling or digital literacy.
- The introduction is too descriptive and reads like a literature review rather than a focused rationale.
- Several citations (e.g., [4–17]) describe consumer behaviour and DTC marketing—these could be summarised more concisely.
- The research gap is stated but not critically justified: it should explicitly explain why PMRR validity matters for clinical or policy implications beyond existing patient-reported outcome measures (PROs).
- Sample size calculation is not well justified—no specific effect size or statistical parameters are shown.
- Only Pearson’s r and ANOVA were used; no adjustment for multiple comparisons or confounding (e.g., age, education).
- The analysis lacks regression or multivariate methods to assess independent predictors.
- No mention of normality testing or missing-data handling.
- There is no assessment of convergent validity metrics (e.g., Cronbach’s α or Bland–Altman comparison between PMRR and TSQM).
- Some tables (e.g., Table 4–5) are very long and not reader-friendly; grouping or simplifying would improve readability.
- No control for confounders
- Gender differences reported but not discussed adequately.
- The authors claim PMRR demonstrates “validity comparable to TSQM,” but the correlations are modest.
- Phrases like “PMRR demonstrated convergent validity” should be toned down unless supported by confirmatory factor analysis or more robust validity tests.
- The claim that WePharm could inform reimbursement decisions or clinical practice lacks empirical backing from this study.
- Several references are dated (>10 years old); inclusion of more recent literature on digital health or online patient reviews (2020–2025) is needed.
Author Response
Reviewer 4
Comments and Suggestions for Authors
- The abstract should mention that it was a convenience sample, which affects generalizability.
|
Response: Thank you very much for your comment. We agree and now state explicitly in the methods from the Abstract that a convenience sample was used and also added at conclusion with expecting future research with improved generalizability.
Revised text Reflected at Abstract (line16-18, 29-32) Methods: A cross-sectional survey using a convenience sample was conducted from February to July 2018 among patients aged 50–80 years recruited from four senior welfare centers and one community pharmacy in Seoul. Conclusions: These findings support the potential utility of online drug review systems as complementary tools for capturing real-world patient experience and informing shared decision-making in clinical practice, and as these findings were from convenience sample further re-search is expected with improving generalizability. |
- Overstates the findings (“demonstrated moderate correlation”) without acknowledging possible bias due to sampling or digital literacy.
|
Response: Thank you for raising this important comment. We agree that there may be possible bias regarding convenience sampling and/or digital literacy. During the online surveillance, we’ve been making effort to train the researchers not to affect the result but only assist the navigating the online survey, but still needed to state the potential bias at the limitation. Description of researcher assistance at ‘Methods’ section, and relevant bias at ‘Limitation’ section has been reflected.
Revised text: Reflected at 2.2. Study Design and Survey Administration (line 123-128) During the PMRR survey, which was administered through the WePharm platform, some older participants experienced initial difficulties accessing the system. Therefore, researchers provided assistance with platform navigation (e.g., registration, locating the drug page) when needed. To minimize potential bias, all researchers were trained to avoid offering guidance or feedback related to the review content itself.
Reflected at 4.4. Limitations and Future Directions (line 386-396) |
- The introduction is too descriptive and reads like a literature review rather than a focused rationale.
- Several citations (e.g., [4–17]) describe consumer behaviour and DTC marketing—these could be summarised more concisely.
|
Response Thank you very much for your suggestion, and incorporating with your 4th comment, we made clearer and concisely for the introduction section.
Revised text: Reflected at 1. Introduction (line 37-48) Patient medication reviews and ratings (PMRR) on online platforms reflect pa-tients’ real-world experiences with drug therapies. These narratives commonly address multiple facets of medication use, including perceived effectiveness, covering effec-tiveness, safety, convenience, and affordability. As first-person accounts, they can en-gage readers and encourage patients to seek information, discuss options with provid-ers, and participate in decisions—behaviors linked to better adherence and outcomes [1–3]. In the consumer markets, product reviews reduce uncertainty and guide choices [4–6]. Similarly, patient experience reviews (PERR) help evaluate healthcare services and inform quality improvement [7–9]. While PMRRs do not determine prescriptions, they may shape preferences and foster dialogue with prescribers [10,11], and can reveal adherence barriers or challenges with specific drugs [12–18]. |
- The research gap is stated but not critically justified: it should explicitly explain why PMRR validity matters for clinical or policy implications beyond existing patient-reported outcome measures (PROs).
|
Response
Revised text: Reflected at 1. Introduction (line 65-73) To address this gap, we developed WePharm (wepharm.snu.ac.kr), a digital plat-form for sharing patient medication experiences. This study aimed to assess the validity of PMRRs collected through WePharm as a proxy for TSQM scores among older adults using antihypertensive medications. Specifically, our objectives were: (1) to examine the correlation between PMRRs and TSQM scores across total and subdomain measures, and (2) to evaluate the content validity of PMRRs in reflecting treatment satisfaction. By examining the strength and direction of these correlations, we sought to determine whether online PMRR can provide a practical and efficient means of assessing treat-ment satisfaction in real-world clinical contexts. |
- Sample size calculation is not well justified—no specific effect size or statistical parameters are shown.
|
Response: Thank you very much for your pointing out. The target sample size was determined to detect a moderate correlation (Pearson r ≈ 0.30) between PMRR and TSQM domains with 80% power, requiring at least 85 participants. Anticipating exclusions and incomplete responses, we aimed to recruit ≥500 individuals. The final analytic sample of 313 exceeded this threshold, ensuring adequate power for correlation analyses. Subgroup comparisons were exploratory so interpreted descriptively. We applied above explanation on the manuscript upon your guidance.
Revised Text: Reflected at 2.2. Study Design and Survey Administration (line 107-111) According to the sample size computation, the target sample size was determined to detect a moderate correlation (Pearson r ≈ 0.30) between PMRR and TSQM domains with 80% power, requiring at least 85 participants. Anticipating exclusions and incom-plete responses, we aimed to recruit ≥ 500 individuals. Subgroup comparisons were exploratory so interpreted descriptively. |
- Only Pearson’s r and ANOVA were used; no adjustment for multiple comparisons or confounding (e.g., age, education).
- The analysis lacks regression or multivariate methods to assess independent predictors.
|
Response: Thank you for this thoughtful two comments and always appreciated to receiving your precious insights. We respect your suggestion to include adjustments for multiple comparisons and confounding factors such as age and education. We would like to carefully explain that, our primary research objective was not to examine the independent effects of demographic variables on satisfaction scores, but rather to assess the convergent validity between PMRR and TSQM domains. Accordingly, we employed correlation analyses and known-groups comparisons, which are standard approaches for evaluating construct alignment between instruments. These methods were pre-specified and aligned with the conceptual framework of the study. While we acknowledge that demographic factors may influence satisfaction, adjusting for them in the primary analyses would have shifted the focus away from our core aim. Instead, we did our best to clearly state the purpose of analysis in the methods and declare that the adjustment for covariance was not applied. Nevertheless, to declare this point out, we stated and described as the limitation. Please advise if there are anything we can do better to follow your insight and guidance.
Revised Text: Reflected at 2.5. Statistical Analysis (line 179-186) For data analysis, SAS 9.4 version statistical program and Excel 2016 were used. Analyses were designed to evaluate convergent validity between PMRR and TSQM domains. Description of PMRR and TSQM by participant characteristics was done using one-way ANOVA (Analysis of Variance). Correlation between domains of PMRR and TSQM was done using Pearson’s r. ANOVA and Pearson’s correlation were used to as-sess domain-level associations and known-group differences, but adjustment for de-mographic covariates was not applied as the study did not aim to estimate independent effects.
Revised text: Reflected at 4.4. Limitations and Future Directions (line 414-420) Finally, this study lacked multivariable adjustment for potential confounders such as comorbidities and polypharmacy. Although subgroup analyses were conducted by pa-tient characteristics for each domain, simultaneous adjustment for multiple factors was not performed, as the primary aim was to assess convergent validity rather than causal associations. Nonetheless, comorbidity burden and the number of concurrent medica-tions may have influenced satisfaction ratings and should be considered in future val-idation studies employing multivariable modeling. |
- No mention of normality testing or missing-data handling.
|
Response For the handling of the missing data, all of the 313 participants has been completed without missing data with PMRR and TSQM for the analysis. If there was some of the missing data from socioeconomic status (e.g. refuse to declare from the participant), it was not included in such of sub-analysis by status. |
- There is no assessment of convergent validity metrics (e.g., Cronbach’s α or Bland–Altman comparison between PMRR and TSQM).
|
Response First, PMRR domains are single-item constructs, which precludes internal consistency measures like Cronbach’s α that require multi-item scales. Our primary aim was to assess convergent validity at the domain level rather than reliability within a scale. Second, Bland–Altman analysis is intended for agreement between two continuous measures of the same construct. In our case, PMRR and TSQM differ conceptually and structurally (Likert-type ratings vs. composite scores), making agreement analysis less meaningful. Instead, we focused on correlation patterns and known-group comparisons, which are standard for evaluating construct alignment in exploratory validity studies. Please give us advice if there are any suggestions for assessment for convergent validity metrics and we will do our best on improving completeness of our study. |
- Some tables (e.g., Table 4–5) are very long and not reader-friendly; grouping or simplifying would improve readability.
|
Response We believe this approach is helpful for usability of demonstration of our research, and we appreciate your understanding, but please give us any advice or guidance if there are any aspects we can do. |
- No control for confounders
|
Response Furthermore, our sample size was sufficiently large to ensure stable estimates for correlation analyses, and the purpose was exploration rather than confirmatory. For transparency, we have acknowledged this limitation in the Discussion section and noted that future studies using predictive modeling frameworks could incorporate multivariable adjustment to examine independent predictors of satisfaction. Nevertheless, to declare this point out, we stated and described as the limitation.
Revised text: Reflected at 4.4. Limitations and Future Directions (line 414-420) Finally, this study lacked multivariable adjustment for potential confounders such as comorbidities and polypharmacy. Although subgroup analyses were conducted by pa-tient characteristics for each domain, simultaneous adjustment for multiple factors was not performed, as the primary aim was to assess convergent validity rather than causal associations. Nonetheless, comorbidity burden and the number of concurrent medica-tions may have influenced satisfaction ratings and should be considered in future val-idation studies employing multivariable modeling. |
- Gender differences reported but not discussed adequately.
|
Response: Thank you for this helpful comment. We agree that the observed gender differences warrant further discussion. We have now added a paragraph in the Discussion section to interpret these findings in light of prior literature. Specifically, we note that female participants reported higher satisfaction in certain domains, which aligns with previous studies showing that women often express greater treatment satisfaction, potentially due to differences in expectations, communication styles, or health-seeking behaviors. We also acknowledge that these patterns may be influenced by cultural and contextual factors, and suggest that future research explore gender-specific drivers of satisfaction in more depth.
Revised text: Reflected at 4.1. Key Study Findings (line 326-339) When interpreted together, a consistent trend emerged: higher scores were reported by older female patients with lower copayments, while lower scores were given by younger, more affluent, and more educated patients with fewer comorbidities. This pattern is consistent with prior research indicating that women may report greater sat-isfaction with healthcare experiences, possibly due to differences in expectations, communication preferences, or engagement with treatment. A large U.S. study demonstrated that older age and female sex were independently associated with higher patient satisfaction across healthcare settings [28]. Similarly, previous systematic re-search reported that age, sex, marital status, and socioeconomic status significantly in-fluence satisfaction, with older patients and women consistently reporting higher sat-isfaction [29]. These findings suggest that gender may play a role in shaping perceived treatment benefit and satisfaction, and highlight the importance of considering so-cio-demographic factors in the interpretation of patient-reported outcomes. Further research is needed to explore the underlying mechanisms and cultural influences that may contribute to these differences. |
- The authors claim PMRR demonstrates “validity comparable to TSQM,” but the correlations are modest.
|
Response
Revised Text: Reflected at 4.2. Convergence Between PMRR and TSQM (line 350-372) PMRR domains showed moderate correlations with corresponding TSQM con-structs. Specifically, PMRR Effectiveness, Side effects, and Easy to take were strongly associated with TSQM Effectiveness, Side Effects, and Convenience, respectively. PMRR Willingness to recommend aligned most closely with TSQM Global Satisfaction. Importantly, PMRR also introduced unique domains—Affordability and Food interaction—not captured by TSQM. As expected, these domains showed weaker correlations due to the lack of conceptual overlap. Given that PMRR domains are single-item constructs, internal consistency metrics and confirmatory factor analysis are not applicable. Therefore, our validity assessment focused on convergent patterns with TSQM domains and known-group differences, providing preliminary evidence of construct alignment rather than structural equivalence. Nevertheless, PMRR domains showed significant correlations with treatment satisfaction, suggesting moderate convergent validity while expanding the scope to include patient concerns highly relevant in real-world practice, such as cost and dietary compatibility. Although no prior studies have directly compared PMRR with TSQM, the TSQM has been widely validated across conditions [19, 25, 27, 31]. Prior work has also shown consistent demographic influences on satisfaction: older age and female sex are generally associated with higher satisfaction, whereas higher education and income are often linked to lower satisfaction, likely due to elevated expectations [28, 29]. Although this study did not include qualitative data, future mixed-methods research incorporating patient interviews or narrative analyses of PMRR content could further elucidate how expectations shape satisfaction across socioeconomic groups. |
- Phrases like “PMRR demonstrated convergent validity” should be toned down unless supported by confirmatory factor analysis or more robust validity tests.
|
Response: Thank you for this important comment. We agree that the phrase “demonstrated convergent validity” may overstate the findings given the absence of confirmatory factor analysis or other structural validity tests. We have revised the manuscript to use more cautious language, such as “PMRR demonstrates moderate convergent validity,” and clarified that our analyses were exploratory and based on correlation and known-groups comparisons. We also note that PMRR domains are single-item constructs, which limits the applicability of internal consistency or factor-based approaches.
Revised Text: Reflected at 4.2. Convergence Between PMRR and TSQM (line 350-372) PMRR domains showed moderate correlations with corresponding TSQM con-structs. Specifically, PMRR Effectiveness, Side effects, and Easy to take were strongly associated with TSQM Effectiveness, Side Effects, and Convenience, respectively. PMRR Willingness to recommend aligned most closely with TSQM Global Satisfaction. Im-portantly, PMRR also introduced unique domains—Affordability and Food interac-tion—not captured by TSQM. As expected, these domains showed weaker correlations due to the lack of conceptual overlap. Given that PMRR domains are single-item constructs, internal consistency metrics and confirmatory factor analysis are not applicable. Therefore, our validity assessment focused on convergent patterns with TSQM domains and known-group differences, providing preliminary evidence of construct alignment rather than structural equivalence. Nevertheless, PMRR domains showed significant correlations with treatment satisfaction, suggesting moderate convergent validity while expanding the scope to include patient concerns highly relevant in real-world practice, such as cost and dietary compatibility. Although no prior studies have directly compared PMRR with TSQM, the TSQM has been widely validated across conditions [19, 25, 27, 31]. Prior work has also shown consistent demographic influences on satisfaction: older age and female sex are generally associated with higher satisfaction, whereas higher education and income are often linked to lower satisfaction, likely due to elevated expectations [28, 29]. Although this study did not include qualitative data, future mixed-methods research incorporating patient interviews or narrative analyses of PMRR content could further elucidate how expectations shape satisfaction across socioeconomic groups. |
- The claim that WePharm could inform reimbursement decisions or clinical practice lacks empirical backing from this study.
|
Response: Thank you for this important comment. We understand it is careful to state supporting decisions and/or clinical practice without direct empirical evidence from WePharm study. We have revised the manuscript to clarify that PMRRs may offer complementary insights into patient experiences, particularly regarding affordability and usability, but that their application in policy or clinical decision-making requires further validation. We have rephrased the relevant statements to avoid overstatement.
Revised Text: Reflected at 4.3. Implications for Practice and Policy (line 373-385) The results indicate that online review platforms such as WePharm can generate meaningful and valid measures of treatment satisfaction, showing moderate conver-gent validity with traditional survey instruments. By incorporating dimensions such as affordability and food interaction, PMRR provides a broader perspective on the patient experience. These insights may support shared decision-making between patients and healthcare providers and help identify barriers to adherence in real-world settings. From a policy perspective, platforms like WePharm could contribute to pa-tient-centered care by surfacing real-world concerns—such as cost burden and regimen usability—that are often underrepresented in conventional PROs. While PMRR data may complement clinical trial evidence and post-marketing surveillance, their use in reimbursement decisions or clinical practice guidelines would require further empirical validation, standardization, and governance. |
- Several references are dated (>10 years old); inclusion of more recent literature on digital health or online patient reviews (2020–2025) is needed.
|
Response |

Round 2
Reviewer 2 Report
Comments and Suggestions for Authors
Authors have addressed all of my comments/suggestions in their revised submission.
Author Response
Thank you very much for you reply, and thank you again for helping us improving our manuscript.
Reviewer 3 Report
Comments and Suggestions for Authors
The authors have substantially improved the manuscript and addressed several earlier concerns. The study is well written and tackles an important topic in patient-centered evaluation of medication satisfaction using an innovative online PMRR platform. The inclusion of affordability and food interaction domains adds real-world relevance.
However, several key issues remain and should be addressed before acceptance.
- Representativeness & Selection Bias
Please include a participant flowchart with numbers screened, excluded, and analyzed, and provide a brief comparison between included and excluded participants (age, sex, etc.). Discuss how exclusion of those unable to identify medications might bias results toward higher health literacy. - Assistance with Online Platform
Report the proportion of participants who received navigation help and compare their scores and correlations with those who did not to evaluate bias. - Statistical Methods for Ordinal Data
Re-analyze key correlations using Spearman’s rho (report both Pearson and Spearman with 95% CIs) and use non-parametric tests for subgroup comparisons. - Confounding Variables
Include at least one multivariable or partial-correlation model adjusting for major covariates (age, education, comorbidity, polypharmacy). - Single-Item PMRR Measures
Provide brief details of pilot or cognitive testing that ensured content validity of single-item domains. If not done, clearly acknowledge this limitation and note the need for future psychometric validation. - Mode & Carryover Effects
Clarify the survey order and acknowledge that the fixed sequence could influence results. - Medication Misclassification
Quantify exclusions due to inability to name medications and discuss how this may have affected validity. - Reporting
Add 95% confidence intervals, indicate how multiple comparisons were handled, and provide a short statement confirming independence from any industry-affiliated authors.
Author Response
*. General reply
Thank you very much for giving us your precious comments and was important to imporve our study. We would like to reply your comments as below, and also revised some sentence and expressions to give more clarity of our research.
Reviewer
The authors have substantially improved the manuscript and addressed several earlier concerns. The study is well written and tackles an important topic in patient-centered evaluation of medication satisfaction using an innovative online PMRR platform. The inclusion of affordability and food interaction domains adds real-world relevance.
However, several key issues remain and should be addressed before acceptance.
1. Representativeness & Selection Bias
Please include a participant flowchart with numbers screened, excluded, and analyzed, and provide a brief comparison between included and excluded participants (age, sex, etc.). Discuss how exclusion of those unable to identify medications might bias results toward higher health literacy.
|
Thank you very much for your precious comment, and we believe this will be important to improve completeness of our study and demonstration.
First, we’ve showed the participant flowchart, but instead of pasted bitmap file we’ve transformed to vector format so expected no loss of image quality. And we also gave more detail on excluded participants with also mentioning at the first part of section 3.1.
Second, respecting on your comment about excluded participants, we are providing brief comparison of baseline characteristics on below (among 217 excluded participants, pilot investigation with 7 participants were not compared on this analysis). Accepting your point out with health literacy and we assume this can be investigated with age, we’ve divided with two exclusion groups with age ineligibility (age over 80 years), and exclusion from other reasons. Fortunately, they have reported TSQM survey, so we are also comparing TSQM results and included in this comparison.
Regarding with selection bias, we have included limitations at 1st review round that low literacy might have possibility to affect bias, and from the further investigation on excluded participants, some of the baseline characteristics may differ (in excluded participants, more proportion of female included, higher age, less proportion of education), but from the TSQM results there was not big difference between groups with FAS and excluded participants. We added information at limitation that excluded participants with survey incompletion tended to have older age and lower educational attainment.
Manuscript changes: Revised Figure 2 Flowchart of Disposition of study participants.
Manuscript changes: Revised at section 3.1. Baseline Characteristics of the Study Population (line 189-192 added) During the study period, 530 participants were enrolled. Of these, 217 participants were excluded — primarily due to survey incompletion (n=182) and age ineligibility (n=91) — resulting in a full analysis set of 313 participants (Figure 2).
Manuscript changes: Revised at section 4.4. Limitations and Future Directions (line 405-407) Also, many participants required assistance with digital platforms, which implies the digital literacy of participants may have influenced the quality of PMRR data, introducing interviewer assistance related biases, although the assistance was limited to navigation. Furthermore, excluded participants with survey incompletion tended to be older and have lower educational attainment [30]. |
2. Assistance with Online Platform
Report the proportion of participants who received navigation help and compare their scores and correlations with those who did not to evaluate bias.
|
We deeply respect and appreciate this important suggestion. Unfortunately, we did not gather the information whether who received or not with the navigation on online review. It was naturally happened in the course of providing webpage URL informing account registration, and some requested to help out where to navigate to record the drug review at the online.
Respecting your comments, we’ve already reflected at the limitation section (line 402-4-6), but please guide us if there are anything we can do to supplement this.
Manuscript change (at 1st round): 4.4. Limitations and Future Directions Also, many participants required assistance with digital platforms, which implies the digital literacy of participants may have influenced the quality of PMRR data, introducing interviewer assistance related biases, although the assistance was limited to navigation. Furthermore, excluded participants with survey incompletion tended to be older and have lower educational attainment [30]. |
3. Statistical Methods for Ordinal Data
Re-analyze key correlations using Spearman’s rho (report both Pearson and Spearman with 95% CIs) and use non-parametric tests for subgroup comparisons.
4. Confounding Variables
Include at least one multivariable or partial-correlation model adjusting for major covariates (age, education, comorbidity, polypharmacy).
|
Thank you very much for your guidance. We performed Pearson and Spearman, and unadjusted model and partial-correlation model, adjusting covariates with sex, age, education, comorbidity, and polypharmacy. The results are demonstrated at Table 5 and section 3.4.
We’ve reflected at manuscript with unadjusted model and partial-correlation model, but not reflected in manuscript for spearman model because it showed the similar results with pearson model. Please advise or guide us if also showing Spearman model and/or 95% confidence interval in manuscript seems needed.
Table 5. Correlation with using Pearson model
Additional analysis. Correlation with using Spearman model.
Manuscript changes: Revised 2.5. Statistical Analysis (line 188-192) Correlation between domains of PMRR and TSQM was done using Pearson’s r, and furthermore performed Pearson partial-correlation with adjusting major participant’s covariates (sex, age, education, comorbidity, polypharmacy).
Manuscript changes: Revised Table 5. Correlation between dimensions of PMRR and TSQM (N=313).
Manuscript changes: Revised 3.4. Correlation Between PMRR and TSQM Domains (line 303-305) Partial correlation model with adjusting covariates (sex, age, education, comorbidity, and polypharmacy) has also been performed with showing similar results. |
5. Single-Item PMRR Measures
Provide brief details of pilot or cognitive testing that ensured content validity of single-item domains. If not done, clearly acknowledge this limitation and note the need for future psychometric validation.
|
Thank you for this insightful suggestion. We did conduct a small pilot prior to fielding with 7 participants at the Seodaemun senior center to assess feasibility of our survey operation and to investigate any of unexpected situation with establishing preventive plan for the researchers. But as you pointed out, we acknowledge that formal psychometric work (e.g., test–retest reliability; additional qualitative cognitive interviewing) was not performed in this study, and we have added this as a limitation with a note on future validation.
Manuscript change: Revised at 4.4. Limitations and Future Directions (line 439-442) Furthermore, because PMRR domains are single-item measures, we did not perform formal psychometric validation (e.g., test–retest reliability or cognitive interviewing) and future studies will undertake these procedures to further establish content validity. |
6. Mode & Carryover Effects
Clarify the survey order and acknowledge that the fixed sequence could influence results.
|
Thank you for the comment.
We have clarified the administration order in the Methods as after written informed consent, participants completed a paper questionnaire capturing participant characteristics and the TSQM (~20 minutes), followed by a brief rest, and then completed the online PMRR drug review on the WePharm platform (~10 minutes). The sequence was fixed (not randomized). If participants encountered difficulties navigating the online PMRR, trained staff provided standardized technical navigation only (e.g., logging in, advancing pages). We did not track assistance or timing at the participant level. We’ve already added to the Limitations that the fixed order and mixed modes (paper TSQM then online PMRR) could introduce order/carryover and mode effects but added more descriptions that this order was not randomized and suggestions for future studies.
Manuscript change: Revised at 2.2. Study Design and Survey Administration (line 144-147) The survey followed a fixed sequence and instrument order was not randomized. After written informed consent, participants completed a paper questionnaire that included sociodemographic/clinical characteristics and the TSQM, followed by a brief rest. Participants then completed the online PMRR drug review on the WePharm platform.
Manuscript change: Revised at 4.4. Limitations and Future Directions (line 443-447) Because instrument order was not randomized and assistance/timing were not tracked, we were unable to quantify these effects. Future studies will counterbalance order, evaluate mode equivalence, and record assistance, device type, and timestamps to assess potential bias. |
7. Medication Misclassification
Quantify exclusions due to inability to name medications and discuss how this may have affected validity.
|
Thank you very much for your point out. The majority of exclusions were caused because of survey incompletion (182 incompletion out of 210 exclusions), and the majority of occurrence was because long taken survey time with multiple survey tools. They performed participant’s characteristics and TSQM first, but dropped out during the rest time prior to perform online survey with PMRR. As ‘named medications’ is gathered at PMRR but not TSQM, there is possibility that some of them might ‘inability to name’ but difficulties are expected to quantify.
For the investigation on excluded participants - in accordance from your 1st comment - where majority of them were incompletion, we figured out that TSQM results were similar with FAS analysis group.
Please advise if there are anything we can do more to improve completion of our study regarding with this comments. |
8. Reporting
Add 95% confidence intervals, indicate how multiple comparisons were handled, and provide a short statement confirming independence from any industry-affiliated authors.
|
Thank you very much for your precious comments.
First of all, we’ve demonstrated 95% confidence intervals in Table 3 and Table 4. We also added median at table 2 to enhance demonstration of distribution.
Second, we’ve stated pairwise comparison by subgroups if significant group differences shown in ANOVA model from PMRR and TSQM (Table 3, 4). If significance shown in pair-wise model, it has been marked with asterisk, daggar, etc.
Third, we’ve enhanced the statement of confirming independence at COI section.
Manuscript change: Revised at 2.5. Statistical Analysis (line 184-185) If there were shown significant group difference, pair-wise comparison by subgroups were also performed.
Manuscript change: Revised at Table 2. Comparison of responses to PMRR and TSQM. Added median.
Manuscript change: Revised at Table 3. PMRR by participant characteristics., Table 4. TSQM by Participant Characteristics. Added 95% confidence interval.
Manuscript change: Conflicts of Interest (line (475-478) D.H.K. is currently an employee of Sanofi Korea, and declare not having conflict of interests with study participated cites. T.Y. is currently an owner of Bulgwang Ildeung Pharmacy but not owned or working at pharmacy at study period, and declare not having conflict of interest with study participated cites. |

Reviewer 4 Report
Comments and Suggestions for Authors
The manuscript is now improved
Author Response
Thank you very much for you reply, and thank you again for helping us improving our manuscript. We've revised the manuscript according to other reviewer, and also improved Figures and expressions to improve more clarity of our research.
